# CRISPR-dCas13d-based deep screening of proximal and distal splicing-regulatory elements

Yocelyn Recinos[1,2], Dmytro Ustianenko[1,2,3], Yow-Tyng Yeh[1,2], Xiaojian Wang[1,2], Martin Jacko[1,2,4], Lekha V. Yesantharao[1,2,5], Qiyang Wu[1,2] & Chaolin Zhang [1,2] ✉

Pre-mRNA splicing, a key process in gene expression, can be therapeutically modulated using various drug modalities, including antisense oligonucleotides (ASOs). However, determining promising targets is hampered by the challenge of systematically mapping splicing-regulatory elements (SREs) in their native sequence context. Here, we use the catalytically inactive CRISPR-*Rfx*Cas13d RNA-targeting system (dCas13d/gRNA) as a programmable platform to bind SREs and modulate splicing by competing against endogenous splicing factors. SpliceRUSH, a high-throughput screening method, was developed to map SREs in any gene of interest using a lentivirus gRNA library that tiles the genetic region, including distal intronic sequences. When applied to *SMN2*, a therapeutic target for spinal muscular atrophy, SpliceRUSH robustly identifies not only known SREs but also a previously unknown distal intronic SRE, which can be targeted to alter exon 7 splicing using either dCas13d/gRNA or ASOs. This technology enables a deeper understanding of splicing regulation with applications for RNA-based drug discovery.

Most mammalian genes have a split gene structure consisting of exons and introns, where introns are typically much longer than exons (median: ~1800 bp and 120 bp, respectively; 17% introns >10 kb). Producing mature mRNA transcripts by precisely removing introns and ligating exons through RNA splicing is a prerequisite for protein translation and the proper function of many noncoding RNAs[1–3]. In addition, over 90% of multi-exon genes undergo alternative splicing to generate multiple transcript and protein isoforms from a single gene[4,5], providing a major source of molecular diversity required for organismal development and function[6].

Splice site recognition and exon inclusion levels are tightly regulated via numerous splicing-regulatory elements (SREs) embedded in the exon and flanking introns. These elements can interact with hundreds of RNA-binding proteins (RBPs) or mediate the formation of local or long-range RNA-secondary structures, thereby either assisting or interfering with the recruitment of the spliceosome[7–9]. Given their

importance for splicing and gene expression, mutations that disrupt SREs are implicated in many diseases, ranging from neurodevelopmental disorders to various cancers[10–12]. Modulating splicing with antisense oligonucleotides (ASOs) and other drug modalities has emerged as a promising therapeutic strategy, particularly for diseases that lack viable targets using traditional approaches[13,14]. So far, ASOs have been successfully applied to trigger the skipping of exons carrying disease-causing mutations (e.g., Duchenne muscular dystrophy[15]), to correct an aberrant disease-causing splicing pattern, and to modulate gene expression by targeting endogenous poison exons that contain in-frame stop codons[16,17]. This strategy has resulted in several FDA-approved drugs and an expanding list of clinical trials (see review by ref. 18).

Despite the exciting progress, a bottleneck of the field is the incomplete understanding of the "splicing code", which is highlighted by the sparse number of annotated functional SREs in the human

[1]Department of Systems Biology, Columbia University, New York, NY 10032, USA. [2]Department of Biochemistry and Molecular Biophysics, Columbia University, New York, NY 10032, USA. [3]Present address: Flagship Pioneering, Cambridge, MA 02142, USA. [4]Present address: Aperture Therapeutics, Inc., San Carlos, CA 94070, USA. [5]Present address: Johns Hopkins University School of Medicine, Baltimore, MD 21205, USA. ✉e-mail: cz2294@columbia.edu

genome, especially when compared to the number of identified epigenetic and transcriptional regulatory elements[19]. To date, most experimental[20–24] and computational methods[25–27] focus on SREs in exonic and proximal intronic sequences (-200 nucleotides from exons or less) and frequently rely on heterologous systems to identify SREs. It is well known that the short and degenerate sequence identity of SREs is generally insufficient to determine their regulatory activity. This is emphasized by the important role that sequence context plays, which is difficult to predict accurately with current computational models[28–30]. Furthermore, distal intronic sequences are under evolutionary constraints related to splicing[31], and individual examples of distal intronic SREs have been reported[32,33], suggesting their relevance and potentially widespread impact on splicing regulation. However, there is currently no effective method to identify distal SREs systematically. For example, ASO drug screenings still rely on expensive and time-consuming "ASO walks" to test the impact of individual ASOs targeting the exon of interest and proximal intronic sequences while omitting distal SREs due to the limited scalability of these screenings.

The class 2, type VI CRISPR-Cas RNA-targeting system, which includes the effector proteins Cas13a-d, X, and Y identified so far, provides a powerful, programmable tool for transcriptome engineering in mammalian cells and animals[34–41]. Targeted cleavage of specific RNA transcripts by the Cas13 protein is achieved by using a guide RNA (gRNA) that consists of a direct repeat (DR) and a 22–30 nucleotide (nt) spacer that is complementary to the target RNA sequence. High-throughput knockdown screenings of genes influencing cellular fitness, targeting circular RNAs and long noncoding RNAs[41–47] have been performed using the Cas13 system and a pooled lentiviral gRNA library. In addition, the catalytically inactive Cas13 (dCas13) has been used as a programmable RNA-binding module, fused to various effector

domains for applications in RNA editing[48], m⁶A modifications[49], modulation of splicing[37,50,51] and polyadenylation[52], live cell RNA imaging[53], and mapping of protein-RNA interactions through proximity ligation[54]. Finally, attempts to modulate splicing or translation[55] by using dCas13/gRNA to compete with endogenous RNA-binding effectors (e.g., RBP) through steric hindrance have yielded results with varying degrees of success. So far, dCas13 applications have been limited to individual gRNAs and targets, with components frequently introduced through transient overexpression. Lentivirus-based high-throughput screening using dCas13 and a pooled gRNA library has yet to be established despite the tremendous potential of this approach, as has been demonstrated by analogous efforts for other CRISPR systems such as Cas9. In this study, we report a dCas13d/gRNA-based high-throughput screening platform, splicing regulation uncovered by steric hindrance (SpliceRUSH), to systematically map proximal and distal SREs for any exon of interest within its native sequence context.

## Results

### Overview of the SpliceRUSH screen

The SpliceRUSH screening platform utilizes *Rfx*Cas13d (CasRx)[37], one of the most widely used Cas13 effectors for transcriptome engineering applications. The nuclease-inactive mutant, d*Rfx*Cas13d (dCas13d), when in complex with a gRNA, binds but does not cleave the target RNA. The binding of the dCas13d/gRNA complex to an SRE in the pre-mRNA antagonizes the cognate splicing factor(s) or RNA-secondary structure(s) that either positively or negatively regulate splicing, resulting in the modulation of splicing through steric hindrance. This mode of action is shared by ASOs, which are similar in length (-15–30 nt) to Cas13 gRNAs (22–30 nt; Fig. 1a). Therefore, based on the splicing modulatory impact of gRNAs targeting specific RNA

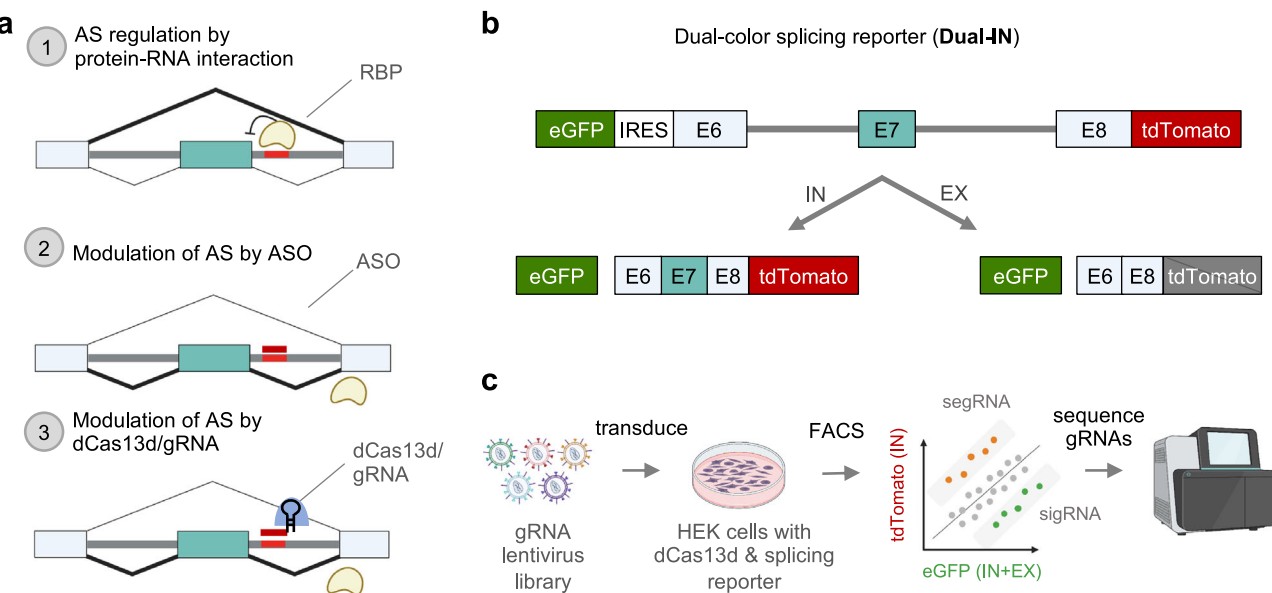

**Fig. 1 | Overview of the SpliceRUSH screening method. a** Schematics showing the use of dCas13d/gRNA as a programmable RNA-binding platform to map both proximal and distal SREs in their native sequence context by binding to target transcripts and competing with endogenous RNA-binding proteins (RBPs) to positively or negatively modulate the transcript's splicing. The example illustrates that blocking an intronic silencer element induces exon inclusion. Since steric hindrance is a shared mechanism between antisense oligonucleotides (ASOs) and dCas13d/gRNA, candidate SREs identified by the screen can be targeted with ASOs. **b** Schematic representation of the Dual-IN *SMN2* splicing reporter with a frame-shifting exon 7 and flanking intronic and exonic regions. This reporter is fused to eGFP and an internal ribosomal entry site (IRES) upstream of exon 6, and tdTomato is downstream of exon 7, so that the construct expresses eGFP and tdTomato fluorescence when the frame-shifting exon is included in the transcript and only

eGFP fluorescence when the frame-shifting exon is excluded. **c** Overview of the lentiviral- based pooled gRNA library screening. Cells stably expressing the dual-color splicing reporter and dCas13d-BFP are transduced with a pooled lentiviral gRNA library that tiles the entirety of the reporter's genetic region. The cells are then sorted based on tdTomato and eGFP fluorescence. The subpopulation of cells with high tdTomato/eGFP ratio (top bin, splicing-enhancing gRNAs, or segRNAs, for the Dual-IN *SMN2* splicing reporter) and low tdTomato/eGFP ratio (bottom bin, splicing-inhibiting gRNAs, or sigRNAs, for the Dual-IN *SMN2* splicing reporter), together with unsorted cells, are collected for gRNA sequencing and bioinformatic analysis. The gRNAs enriched in the top and bottom bins are expected to target SREs that modulate the splicing of the frame-shifting exon. The schematics in panels **a** and **c** were generated using BioRender.

sequences, this approach enables the mapping of SREs in their native sequence context and the identification of regions that can be targeted by dCas13d/gRNA or ASOs to modulate splicing for therapeutics.

To develop SpliceRUSH, we used *SMN2* exon 7 as a model splicing system, as it has undergone extensive screening for SREs due to its role as a therapeutic target for spinal muscular atrophy (SMA)[56]. These studies include ASO screenings, which led to the development of the FDA-approved ASO therapeutic Spinraza/nusinersen, the first efficacious treatment for SMA. SMA is caused by a deletion or mutation in the *Survival Motor Neuron 1* (*SMN1*) gene, resulting in a loss of the SMN protein. While humans have a nearly identical paralogous *SMN2* gene, its expression does not rescue the defects as the primary splicing isoform excludes exon 7, producing an unstable and non-functional protein[17,57]. Nusinersen targets an intronic splicing silencer immediately downstream of *SMN2* exon 7, named ISS-N1, and prevents hnRNP A1/A2 from binding and suppressing exon 7 inclusion[58]. This results in the modulation of splicing that leads to the inclusion of exon 7 in the final transcript and rescues SMN protein production in SMA patients[59]. Despite its remarkable success, a significant effort was required to identify the optimal SRE to target, mainly led by the tiling of ASOs at different step sizes (ASO walk and micro-walk) along *SMN2* exon 7 and flanking intronic regions[60–63].

To evaluate the splicing-modulatory impact of gRNAs targeting different sequences in a high-throughput manner, we used a reporter system that expresses fluorescent proteins eGFP and tdTomato as readouts to indicate the abundance of a specific splicing isoform. Specifically, we used an *SMN2* minigene that contains exon 7 along with its flanking introns and exons (exons 6 and 8)[64] and generated dual-color splicing reporters with slight but important modifications in the included *SMN2* sequence. The dual-color reporters express eGFP constitutively as a measure of reporter expression level. tdTomato expression is contingent on the inclusion or exclusion of the frame-shifting cassette exon 7, which depends on the reporter construct. For method development, we initially used the Dual-IN version of the splicing reporter, in which tdTomato is in-frame and translated when exon 7 is included in the transcript (Fig. 1b and Supplementary Fig. 1a). Inversely, for later screens, we used the Dual-EX reporter that expresses tdTomato when exon 7 is excluded from the final transcript (Fig. 4a, b below). We note that similar fluorescent splicing reporters were previously used for high-throughput screening of splicing regulators or splicing-disrupting mutations[21,65,66].

To perform the SpliceRUSH screen, we generated clonal cell lines that stably express dCas13d and the inducible dual-color splicing reporter. A pooled lentiviral gRNA library, designed to tile the splicing reporter at 1-nt step, is transduced into the stable cell line. The cells are sorted using fluorescence-activated cell sorting (FACS) into separate bins based on the ratio of tdTomato to eGFP fluorescence. The enriched gRNAs pertaining to the specific bins with high or low tdTomato/eGFP ratio are identified by deep sequencing and bioinformatics analysis (Fig. 1c). With the Dual-IN reporter, we expect that splicing-enhancing gRNAs (segRNAs) that increase exon 7 inclusion will be enriched in the top bin with a high tdTomato/eGFP ratio, whereas splicing-inhibiting gRNAs (sigRNAs) that decrease exon 7 inclusion will be enriched in the bottom bin with low tdTomato/eGFP ratio. The inverse result is expected for screens using the Dual-EX reporter.

### Specific splicing activation and repression through transient dCas13d/gRNA expression

To confirm that the Dual-IN *SMN2* splicing reporter recapitulates splicing regulation of the endogenous *SMN2* gene and the original minigene used in prior studies[60–63], we transfected the reporter into HEK293T cells followed by RT-PCR and fluorescence analysis, a Dual-IN *SMN1* splicing reporter was included for comparison.

In this experiment, exon 7 was constitutively included in *SMN1*, and the inclusion level was much lower in *SMN2* (percent spliced in or PSI = 100% vs. 49%), as expected (Supplementary Fig. 1b, c). In addition, co-transfection of the Dual-IN *SMN2* splicing reporter with the nusinersen ASO targeting ISS-N1 at varying concentrations led to an increase of exon 7 inclusion in a dose-dependent manner (Fig. 2a, b), consistent with previous results[63]. These data suggest that the Dual-IN *SMN2* splicing reporter is suitable for screening SREs in the *SMN2* gene.

Several recent studies have used dCas13d/gRNA without fused effector domains to modulate splicing, but the effectiveness of this approach remains inconclusive, especially when used for splicing activation[37,50,51]. To confirm that the dCas13d/gRNA system can modulate splicing both positively and negatively and to compare the efficacy of dCas13d/gRNA- and ASO-mediated splicing modulation, we designed multiple gRNAs to target previously identified SREs exhibiting varying regulatory effects on *SMN2* exon 7 when blocked with ASOs (Fig. 2a). HEK293T cells were co-transfected with the plasmids encoding for the Dual-IN *SMN2* splicing reporter, NLS-dCas13d-BFP (NLS=nuclear localization signal), and individual gRNAs, followed by RT-PCR analysis. A gRNA that binds to exon 7 from positions 22 to 40 (E[22,40]) overlaps with an exonic splicing enhancer recognized by the splicing factor Tra2 (refs. 67,68). Targeting this region with ASOs (E[21, 35] and E[26, 40]) was previously shown to reduce exon 7 inclusion[62]. As expected, co-transfection of this gRNA with dCas13d strongly reduced exon inclusion from ~40% to 12% ($p < 0.001$, $n \geq 3$ per group, t-test) (Fig. 2c). Additionally, the ISS-N1 element was targeted with six individually transfected gRNAs (Fig. 2a). Among them, four gRNAs significantly enhanced exon inclusion compared to the controls (PSI = 52–73% vs. 40%; $p < 0.05$ for all tests, $n \geq 3$ per group, t-test), and the magnitude of splicing change is comparable to what was previously reported when ASOs were used to target the same element (Fig. 2c). Two other gRNAs (D[4, 25] and D[8, 29]) reduced exon 7 splicing, likely due to the immediate proximity of the targeted sequences to the 5′ splice site, which may have resulted in the interference of U1 binding and splice site recognition. Finally, the two gRNAs targeting the 3′ and 5′ splice sites, A[−15, 7] and D[−7, 15], also reduced exon 7 inclusion. However, the effects were more moderate than the gRNA E[22, 40] targeting the exonic splicing enhancer.

We found that targeting specific SREs by coexpression of dCas13d and gRNA is required for splicing modulation. Transfection of dCas13d with a non-targeting (NT) gRNA does not alter exon 7 splicing. Moreover, cells treated with E[22,40] or D[9,30] gRNAs without dCas13d also showed similar exon inclusion levels compared to the controls. Our results suggest that dCas13d/gRNA can target specific SREs and effectively modulate splicing positively and negatively through steric hindrance.

Next, we tested whether tdTomato/eGFP fluorescence accurately reports dCas13d/gRNA-mediated splicing changes. To accomplish this, we used flow cytometry to measure the changes in tdTomato relative to eGFP fluorescence intensity after co-transfection of the Dual-IN *SMN2* splicing reporter, dCas13d, and gRNA. For this analysis, we focused on cells expressing dCas13d-BFP (Supplementary Fig. 2). As expected, the tdTomato fluorescence intensity decreased upon transfection of the sigRNA E[22,40] and increased upon transfection of the segRNA D[9,30], as compared to the NT gRNA control (Fig. 2d, top). To perform a more quantitative comparison, we estimated the exon inclusion level based on the ratio of tdTomato to eGFP on a $\log_2$ scale, i.e., $\log_2(\text{PSI}) = \log_2(\text{tdTomato}) - \log_2(\text{eGFP}) + c$, as visualized by an MA-contour plot (c is a constant). Compared to NT control, we observed a higher PSI for the samples treated with segRNAs and a lower PSI for the samples treated with sigRNAs (Fig. 2d bottom and Supplementary Fig. 3). We normalized the PSI value of

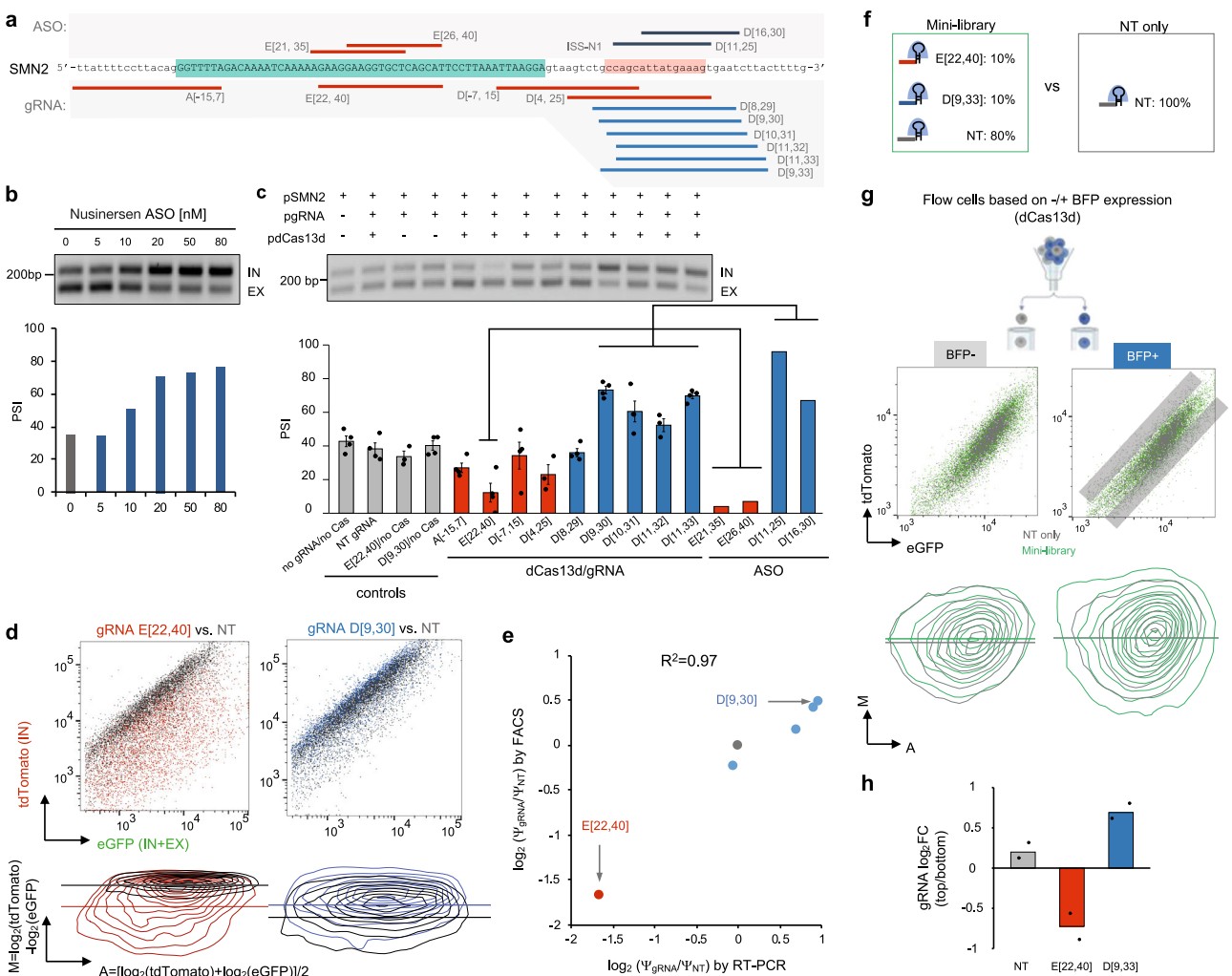

**Fig. 2 | The effect of dCas13d/gRNA complex on splicing modulation by steric hindrance. a** Schematic representation of gRNAs (bottom) and ASOs (top) targeting known SREs in the Dual-IN *SMN2* pre-mRNA, including the intronic splicing silencer element (ISS-N1) targeted by the nusinersen ASO. **b** Exon 7 inclusion shows a dosage-dependent increase upon treatment of the nusinersen ASO. A gel image of splicing products from RT-PCR analysis is shown at the top, and the quantification of exon inclusion levels (percent spliced in or Ψ) is shown at the bottom (*N* = 1). **c–e** Testing dCas13d/gRNA effect on splicing modulation through transient expression. **c** RT-PCR analysis of *SMN2* exon 7 inclusion levels after HEK293T cells were co-transfected with Dual-IN *SMN2* splicing reporter (pSMN2), dCas13d-BFP (pdCas13d), and selected individual gRNAs (pgRNA) is shown. Various controls (first four lanes) were also included for comparison. Quantification shows the average of three or more replicates from independent gRNA transfections with an error bar representing the standard error of the mean (SEM). Published RT-PCR data from ASO-treated cells[64] is plotted on the right, with gRNAs targeting similar sequences highlighted. **d** Fluorescence-based splicing readouts in cells co-transfected with Dual-IN *SMN2*, dCas13d-BFP, and gRNA. Datasets contain ≥5,000 BFP-positive events per sample. Data from splicing-inhibiting gRNA (E[22,40]) and splicing-enhancing gRNA (D[9,30]) are shown as two representatives (see Supplementary Fig. 3 for additional examples). For each gRNA, the FACS plot is shown on the top. The tdTomato and eGFP fluorescence intensities were used to derive an

MA-contour plot, which is shown at the bottom. In the MA-contour plot, the y-axis reflects the exon inclusion level. Note the shift of cell populations on the y-axis when treated with the sigRNA E[22,40] (red) or the segRNA D[9,30] (blue), in comparison with cells treated with a non-targeting gRNA (gray). **e** Correlation of relative *SMN2* exon 7 inclusion level in the log₂ fold change scale in cells treated with different gRNAs vs. non-targeting control as quantified by FACS and RT-PCR. **f–h** Testing a mini-library in a cell line with a DOX-inducible Dual-IN *SMN2* and stable expression of dCas13d-BFP. **f** The gRNA composition of the mini-library and NT only controls. **g** The mini-library or NT control lentivirus was transduced into the Dual-IN *SMN2*/dCas13d-BFP cell line, and cells were fractionated by FACS into BFP-positive (dCas13d-expressing) and BFP-negative populations. Cells in each population were further sorted based on eGFP and tdTomato fluorescence. The FACS plot (top) shows tdTomato vs. eGFP intensity for cells transduced with the mini-library (dark green) overlaid on cells transduced with the NT control gRNA (gray). MA-contour plots are shown at the bottom, with cells transduced with the mini-library (green) overlaid on cells transduced with NT control (gray). For BFP-positive cells transduced with the mini-library, the sub-populations with high or low tdTomato vs. eGFP (i.e., top 5% and bottom 5% bins highlighted by the gray boxes) were collected for gRNA sequencing. **h** The enrichment of the two gRNAs and the NT control in the top vs. bottom bin, log₂ (fold change), was calculated from two replicates of independent gRNA transductions.

each sample relative to the NT control so that the estimated fold changes based on fluorescence are directly comparable to those estimated by RT-PCR. The results from these two quantification methods are highly correlated ($R^2$ = 0.97; Fig. 2e).

Several studies indicated that gRNA length may affect the efficiency of Cas13d-mediated target RNA cleavage[42,44,46,69]. In our initial transfections, we used a 22-nt gRNA, D[9,30], to target the ISS-N1

region as a positive segRNA control. We tested if a larger-sized gRNA targeting the ISS-N1 element would be more potent in splicing modulation. Indeed, a 25-nt gRNA, D[9,33], resulted in a greater increase in exon inclusion, as reflected in both RT-PCR and flow cytometry (Supplementary Fig. 4). For further tool optimizations, we used the gRNAs E[22,40] and D[9,33] as they most drastically modulate exon 7 splicing.

## Splicing activation and repression using lentiviral delivery of gRNAs

Upon confirming the ability of the dCas13d/gRNA system to modulate splicing in both directions, we prepared the tools necessary for the high-throughput screening of SREs using the Dual-IN *SMN2* splicing reporter and a pooled lentiviral gRNA library (Supplementary Figs. 5–8).

To obtain a more homogenous expression of the splicing reporter and an improved control of the stoichiometry of the reporter vs. dCas13d/gRNA expression, we first established a monoclonal HEK293 Flp-In T-REx cell line expressing the doxycycline (DOX)-inducible Dual-IN *SMN2* splicing reporter. DOX-dependent expression of the reporter was confirmed by flow cytometry and RT-PCR (Supplementary Fig. 5). For the other components, we initially attempted to introduce both dCas13d-BFP and gRNA using a one-system lentiviral vector but obtained a very low lentivirus packaging efficiency (Supplementary Fig. 6). We then decided to stably express dCas13d-BFP in the Dual-IN *SMN2* reporter cell line via lentiviral transduction and selection (referred to as the Dual-IN *SMN2*/dCas13d-BFP cell line; Supplementary Fig. 7). This approach also allowed us to optimize the dCas13d-BFP lentiviral concentration during cell transduction and obtain clones with high dCas13d-BFP expression (Supplementary Fig. 8).

To facilitate screening with a pooled lentiviral gRNA library, we evaluated whether gRNAs transduced at a low MOI (i.e., a single viral particle transduction per cell) can sufficiently provide effective splicing modulation. The lentivirus encoding for either sigRNA E[22,40] or segRNA D[9,33] was individually transduced to the Dual-IN *SMN2*/dCas13d-BFP cell line at MOI = 0.3 or 3 (>3 gRNA copies per cell). Consistent with our transfection experiments, clear shifts of tdTomato intensities relative to eGFP were observed in the expected directions with a low MOI (Supplementary Fig. 9), although the splicing-modulatory effects are stronger at the higher gRNA MOI (Supplementary Fig. 10). We noted that a subset of cells no longer expressed BFP after expansion, likely due to the silencing of the dCas13d-BFP locus. We leveraged this observation to fractionate cells with high, medium, and no dCas13d-BFP expression and examined dCas13d/gRNA-mediated splicing modulation in each fraction. This experiment confirmed that both high and intermediate levels of dCas13d-BFP expression can effectively modulate splicing in our system (Supplementary Fig. 11).

With all these components, we further tested a mini-library using a mix of three gRNAs (i.e., NT control, E[22,40], and D[9,33]) in an 8:1:1 ratio when packaging the lentivirus; cells transduced with NT gRNA only were used as a control for comparison (Fig. 2f). The proportion of each gRNA in the mini-library was intended to reflect an assumption that a majority of pre-mRNA sequences targeted by gRNAs in an actual screening library may not have a drastic impact on splicing. The mini-library was transduced into our Dual-IN *SMN2*/dCas13d-BFP cell line at a low MOI (MOI = 0.3). Following selection with puromycin and induction with DOX, cells were sorted by FACS based on the ratio of tdTomato to eGFP fluorescence (Fig. 2g), with top and bottom bins collected (approximately 5% of all BFP-positive cells in each bin; Fig. 2g gray boxes). We analyzed the tdTomato/eGFP ratio in dCas13d-BFP-positive cells while using BFP-negative cells as an internal control (Supplementary Fig. 12). For BFP-positive cells, MA plot analysis revealed an increased range of tdTomato intensity relative to eGFP in both directions upon transduction with the mini-library compared to the NT control, suggesting both an increase and decrease in exon 7 inclusion in different subsets of cells. Importantly, this effect was not observed in dCas13d-BFP-negative cells (Fig. 2g).

The genomic DNA from the collected BFP-positive cells pertaining to bins with high and low tdTomato/eGFP ratio was then used to amplify the gRNA cassette, followed by deep sequencing to determine gRNA enrichment. As expected, we observed that E[22, 40] and D[9, 33] gRNAs were enriched in the bottom and top bins, respectively, while the NT gRNA had a similar representation in the two bins (Fig. 2h). This analysis suggests the feasibility of our system to detect gRNAs that modulate splicing in both directions through high-throughput screening of a pooled gRNA library.

## Pooled gRNA library screening using the Dual-IN *SMN2* reporter

To systematically screen for SREs that impact *SMN2* exon 7 splicing, we designed a gRNA library that tiles all possible pre-mRNA positions at a 1-nt step from the upstream exon 6 to the downstream exon 8 of the Dual-IN *SMN2* splicing reporter (Fig. 3a). In total, the library included 814 on-target 22-nt gRNAs, 811 on-target 25-nt gRNAs, and 312 NT gRNAs as controls (*N* = 1937 in total). The library of gRNA sequences was array-synthesized and cloned into the gRNA lentiviral vector for packaging as a pooled gRNA library.

The lentiviral gRNA library was transduced into the Dual-IN *SMN2*/dCas13d cell line at a low MOI (MOI = 0.3). Cells were fractionated based on BFP expression, and then BFP-positive cells in the top 5% and bottom 5% bins based on the tdTomato/eGFP ratio were collected to quantify gRNA enrichment in each bin by deep sequencing (Supplementary Figs. 12 and 13). A Z-score reflecting the enrichment of each gRNA in the top or bottom bins was estimated by comparison with unsorted cells and with each other to identify the splicing enhancing and silencing gRNAs (Methods).

We performed two independent transductions, and the two replicates showed highly reproducible results. With an initial focus on 25-nt gRNAs, we found that a subset of on-target gRNAs is dramatically enriched or depleted in tdTomato high vs. low cells as indicated by extreme Z-scores (Fig. 3b, c), which are absent for non-targeting gRNAs (Fig. 3d). The screen correctly identified gRNAs targeting regions of known SREs, including the exonic region targeted by E[21,45] and the downstream ISS-N1 region targeted by D[10, 34] (Fig. 3b).

Importantly, our screens also discovered sigRNAs in a more distal region of the downstream intron (~330 nt from the 5′ splice site, gRNA D[333, 357] is a representative), suggesting that a robust splicing enhancer was being targeted (Fig. 3b). This element, which we denote as ISE-D1, was missed in previous studies likely due to its large distance from exon 7. To validate this element, we transfected the gRNA D[333,357] together with dCas13d and the Dual-IN *SMN2* splicing reporter, using gRNAs NT, E[21,45], and D[10, 34] for comparison. This experiment confirmed that gRNA D[333,357] strongly reduced exon 7 inclusion (from 70% to 19.5%), validating the screen results, while gRNAs E[21, 45] and D[10, 34] also modulated exon 7 splicing as expected ($p < 0.01$ for all gRNAs in comparison to NT control, $n = 2$, t-test; Fig. 3e). Similar results were obtained for 22-nt gRNAs, albeit with generally a more moderate splicing-modulatory effect (Supplementary Fig. 14).

## Pooled gRNA library screening using the Dual-EX *SMN2* reporter

With the Dual-IN *SMN2* reporter, we observed a more moderate enrichment of segRNAs in the top bin compared to the enrichment of the sigRNAs in the bottom bin. We reasoned that this could be due to the relatively high tdTomato intensity at the baseline exon 7 inclusion level and the potential saturation of the signal upon further increase in exon inclusion. To overcome this limitation, we generated and validated a Dual-EX *SMN2* splicing reporter, which expresses tdTomato upon exon 7 skipping. With this reporter, we expect segRNAs to be enriched in the low tdTomato/eGFP bin while sigRNAs are enriched in the high tdTomato/eGFP bin (Fig. 4a, b, Supplementary Figs. 1a and 15).

After generating a monoclonal cell line with stable expression of dCas13d-BFP and the DOX-inducible Dual-EX *SMN2* reporter, denoted by Dual-EX *SMN2*/dCas13d-BFP cell line, we transduced it with the same lentiviral gRNA library, followed by the same screening procedure as described above (Supplementary Figs. 16 and 17). Again, we found that the two replicate screens from independent transductions produced highly reproducible and specific results, similar to screens using the

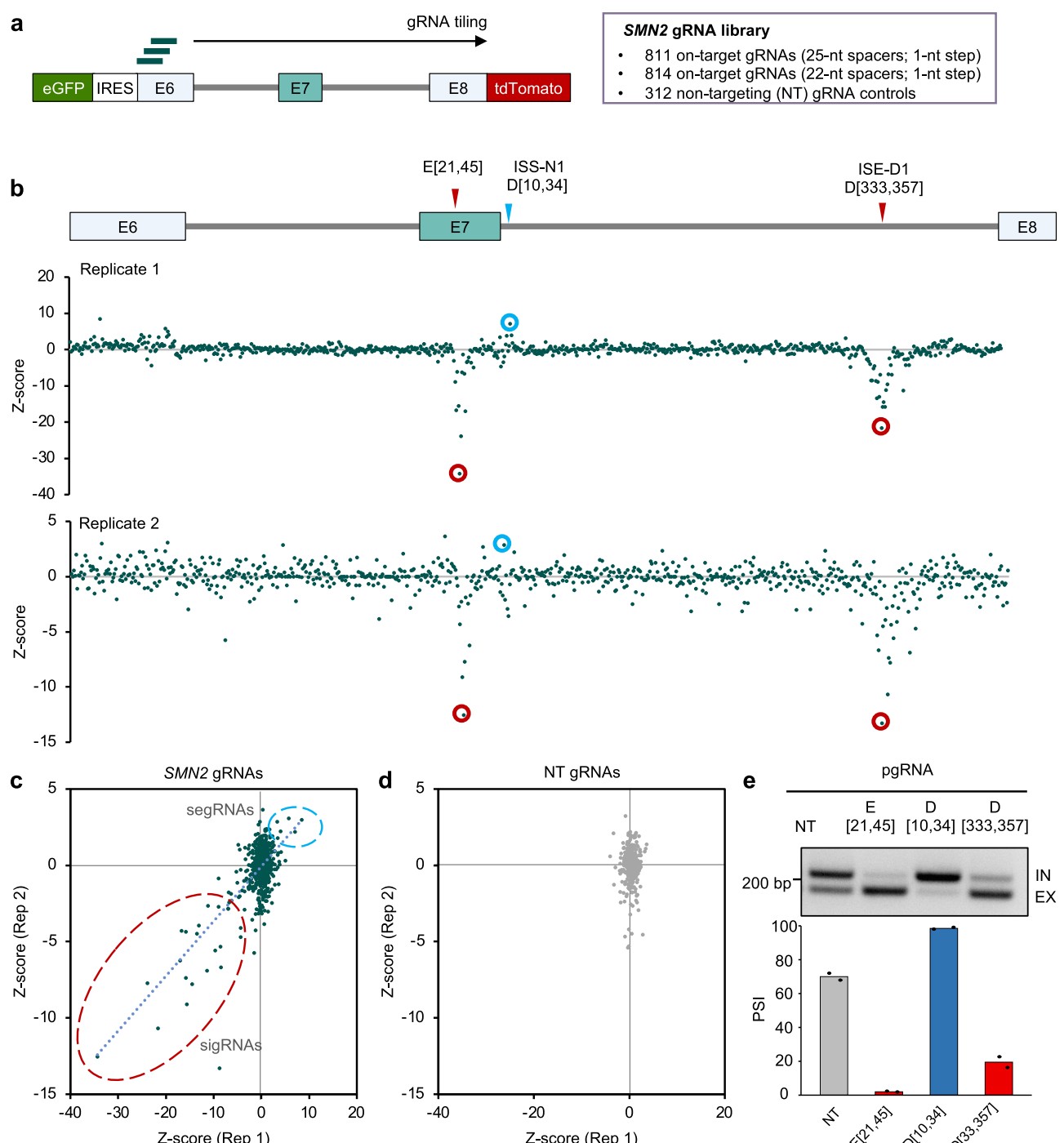

**Fig. 3 | Pooled gRNA library screening using the Dual-IN SMN2 splicing reporter identified a distal intronic SRE. a** Schematic showing the *SMN2* gRNA library consisting of 25-nt and 22-nt spacers tiling the dual color splicing reporter and non-targeting (NT) control gRNAs. **b** gRNA enrichment, as measured by a Z-score comparing cells in the top and bottom bins sorted by tdTomato/eGFP ratio to the unsorted cells, is plotted against the *SMN2* position (the first base pair corresponding to the 3′ end of the gRNA spacer; the same below) for two independent replicates. Results from 25-nt gRNAs are shown. Note that a positive Z-score reflects splicing activation, while a negative Z-score reflects splicing inhibition.

Representative gRNAs targeting known SREs (E[21, 45] and D[10, 34]) and a previously unknown splicing enhancer in downstream intron ISE-D1 (D[333,357]) are highlighted. Correlation of Z-score for on-target (**c**) and NT (**d**) gRNAs between replicates. gRNAs inhibiting E7 splicing are circled in red, and gRNAs enhancing E7 splicing are circled in light blue. **e** Validation of the downstream intronic SRE by transfection of the *SMN2* splicing reporter, dCas13d, and individual gRNA. sigRNA E[21, 45] and segRNA D[10, 34] were also included for comparison. A representative gel image of RT-PCR products is shown at the top, and the quantification of exon inclusion is shown at the bottom.

Dual-IN splicing reporter (Fig. 4c–e and Supplementary Fig. 18). sigR-NAs targeting the exonic splicing enhancer (E[21,45]) and the distal splicing enhancer ISE-D1 that we identified above (D[329, 353]) are now strongly enriched in the top bin, while segRNA D[10, 34] targeting the

ISS-N1 element is enriched in the bottom bin, directions consistent with our expectations. Notably, the magnitude of enrichment for segRNA D[10, 34] targeting the ISS-N1 element improved compared to results from the screens using the Dual-IN splicing reporter.

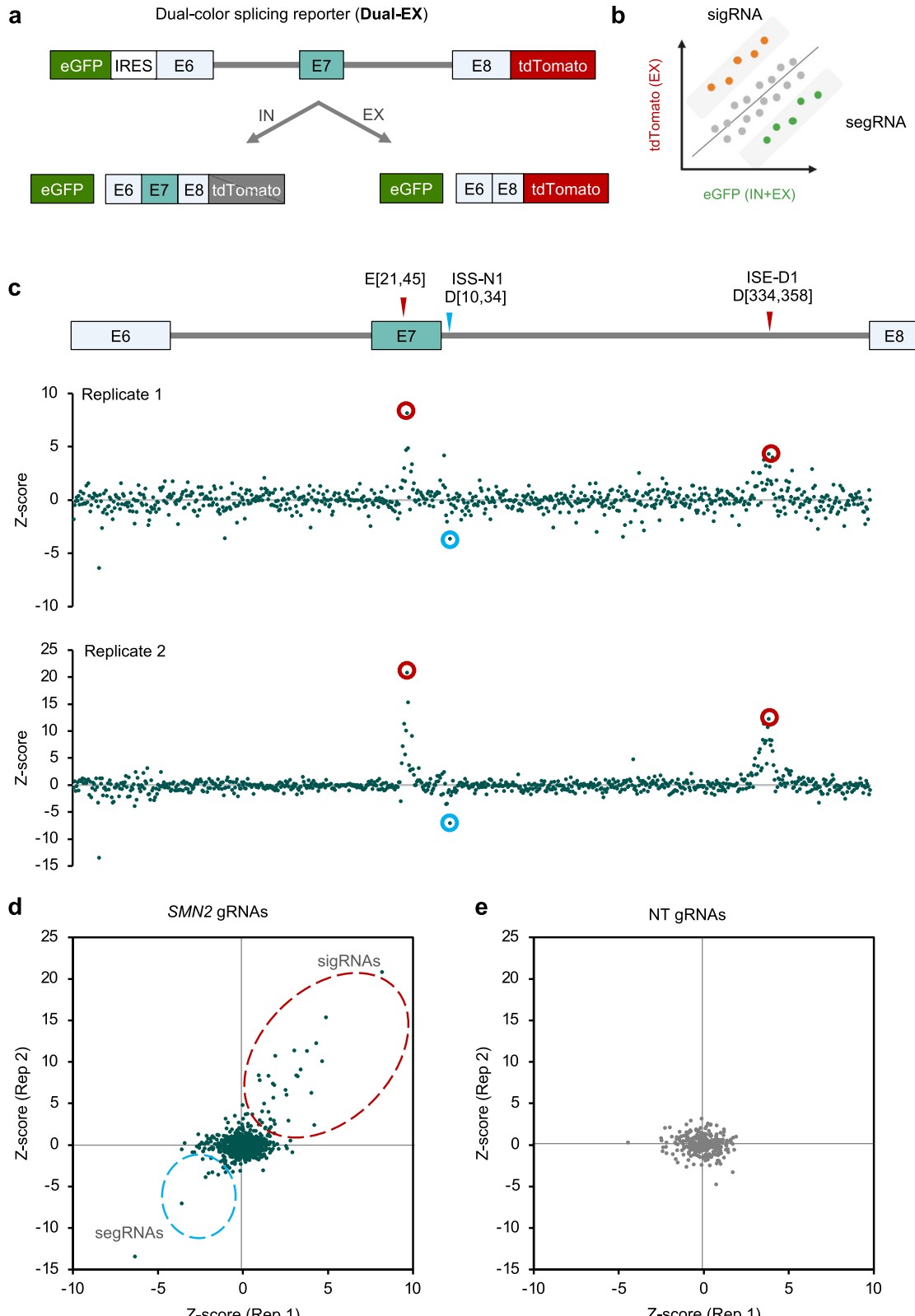

### Effect of gRNA spacer length on splicing modulation

Including 22-nt and 25-nt gRNAs in our pooled gRNA library provided a unique opportunity to evaluate the impact of spacer length on splicing modulation systematically. We first combined all four replicate screens using either Dual-IN or Dual-EX splicing reporters to obtain a final Z-score for each gRNA as a measure of their splicing-modulatory impact (Fig. 5a). As expected, the final Z-scores show a higher signal-to-noise ratio, than the Z-scores from individual replicates, which is particularly clear for gRNAs targeting the ISS-N1 element. We next directly compared the Z-scores of 25-nt gRNAs with those of 22-nt gRNAs. Since each 25-nt gRNA covers four 22-nt gRNAs, we performed separate comparisons for 25-nt vs. 22-nt gRNA pairs with different shifts. Overall, gRNAs with 25-nt spacer have a stronger effect in splicing modulation compared to matching gRNAs with 22-nt spacer (Fig. 5b).

**Fig. 4 | Pooled gRNA library screening using the Dual-EX SMN2 splicing reporter replicates Dual-IN reporter screening results. a** Schematic showing the Dual-EX *SMN2* splicing reporter, which expresses eGFP and tdTomato fluorescence when the frame-shifting exon E7 is excluded and expresses eGFP only when E7 is included. **b** With the Dual-EX reporter, cells containing sigRNAs are expected in the top bin of the tdTomato/eGFP ratio, while cells containing segRNAs are expected in the bottom bin. **c** gRNA enrichment, as measured by a Z-score comparing cells in the top and bottom bins sorted by tdTomato/eGFP ratio, as compared to the unsorted cells, is plotted against the *SMN2* position for two independent replicates. Results from 25-nt gRNAs are shown. Note that a positive Z-score reflects splicing inhibition, while a negative Z-score reflects splicing activation, in contrast to results from the Dual-IN reporter. Representative gRNAs targeting known SREs and the previously unknown splicing enhancer in downstream intron are highlighted. Correlation of Z-score for on-target (**d**) and NT (**e**) gRNAs between replicates. gRNAs inhibiting E7 splicing are circled in red, and gRNAs enhancing E7 splicing are circled in light blue.

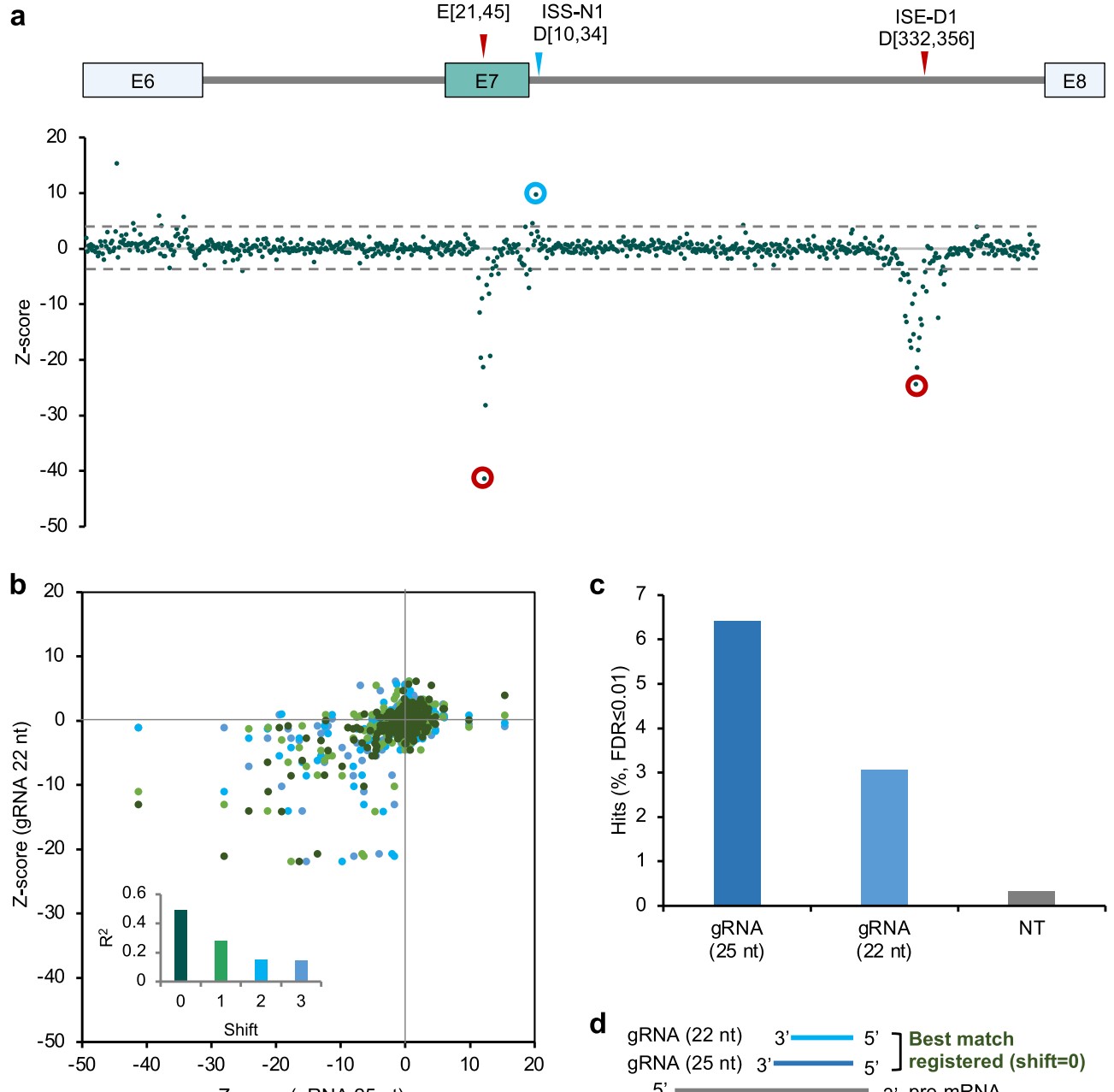

**Fig. 5 | The impact of gRNA spacer length on splicing modulation.**
**a** gRNA enrichment, measured by a Z-score combining all replicate screens using Dual-IN and Dual-EX splicing reporters, is plotted against the *SMN2* position. Results from 25-nt gRNAs are shown. Note that a positive Z-score reflects splicing activation, while a negative Z-score reflects splicing inhibition. Representative gRNAs targeting known SREs and the previously unknown splicing enhancer in downstream intron are highlighted.
**b** Correlation of Z-scores from 25-nt gRNA with 22-nt gRNAs. Each 25-nt gRNA is compared with four different 22-nt gRNAs that fully cover it at different starting positions. Shift 0 is defined when the compared gRNAs have the same 5′ end, while shifts 1, 2, and 3 indicate when the 22-nt gRNA is shifted for that corresponding number of nucleotides towards the 3′ end (i.e., Shift 3 indicates the case when the two compared gRNA coincide at the 3′ end). The inset shows the squared correlation ($R^2$) for each shift.
**c** The percentage of statistically significant hits (FDR ≤ 0.01) for 25-nt and 22-nt on-target gRNAs and NT controls. **d** Illustration showing that the splicing modulatory effect of 25-nt gRNAs best correlates with matching 22-nt gRNAs with the same 5′ end.

In total, 6.4% (52/811) of 25-nt gRNAs are statistically significant for splicing modulation, as compared to 3.1% (25/814) of 22-nt gRNAs, and 0.3% (1/312) of NT controls (FDR < 0.01; Fig. 5). Interestingly, we found that the splicing modulatory effects of 25-nt gRNAs are most correlated with 22-nt gRNAs that have a matching 5′ end (i.e., shift = 0), and least correlated with 22-nt gRNAs that have a matching 3′ end (shift = 3; Fig. 5b inset and d, and Supplementary Fig. 19). This observation is likely because the 5′ end of the gRNA spacer, together with the upstream stem-loop structure, defines the positioning of the dCas13d/gRNA binding footprint on the target RNA.

### Targeting the distal intronic splicing enhancer ISE-D1 using dCas13d/gRNA or ASOs reduces exon 7 inclusion of the endogenous *SMN2* gene

We next tested whether the distal intronic splicing enhancer ISE-D1 identified by the high-throughput screens can be targeted using either dCas13d/gRNA or ASO to modulate exon 7 splicing of the endogenous *SMN2* gene. We first targeted the element by co-transfecting the gRNA D[333,357] with dCas13d. gRNAs targeting E[21,45] and D[10, 34] were also included in this experiment for comparison. As expected, gRNA D[333,357] strongly reduced the inclusion of endogenous *SMN2* exon 7 from 74% to 9%, a magnitude of change comparable to the result when targeted using the sigRNA E[21, 45] (Fig. 6a, b). Quantification of endogenous *SMN1/2* splicing by RNA-seq confirmed that reduction of exon 7 inclusion by gRNA D[333,357] is specific. Treated cells do not have global splicing or gene expression changes (Supplementary

Fig. 20). To confirm whether similar modulatory effects can be achieved using ASOs, we designed several ASOs targeting the same regions (Fig. 6a). After transfections of the ASOs individually into HEK293T cells, *SMN2* exon 7 inclusion was reduced from 81% to 36–48.5% when ISE-D1 is targeted, and ASOs targeting previously known SREs also worked as expected (*p* < 0.01 for all gRNAs in comparison to no gRNA or NT control, *n* ≥ 2, t-test; Fig. 6c). These results indicate that SREs identified as target candidates through our gRNA library screening can indeed be targeted with ASOs to modulate splicing of endogenous genes. This notion is further supported by the observation that gRNA D[10, 34], which targets ISS-N1, activated exon 7 inclusion in the endogenous *SMN2* gene to a similar degree as matching ASOs (Fig. 6b, c).

## Discussion

This study establishes the use of the dCas13d/gRNA system as a programmable RNA-binding platform for high-throughput, deep, and unbiased screening of SREs in their native sequence context. Currently, potential SREs recognized by specific RBPs can be mapped systematically through an "RBP-centric" approach using crosslinking and immunoprecipitation followed by high-throughput sequencing (HITS-CLIP or CLIP-seq)[70,71]. While CLIP experiments have been performed to map binding footprints for hundreds of RBPs (e.g., ref. [72]), such maps are still sparse and limited to certain conditions, such as few cell lines, and the role of the mapped RBP binding sites in splicing regulation has to be further validated using independent assays. To the

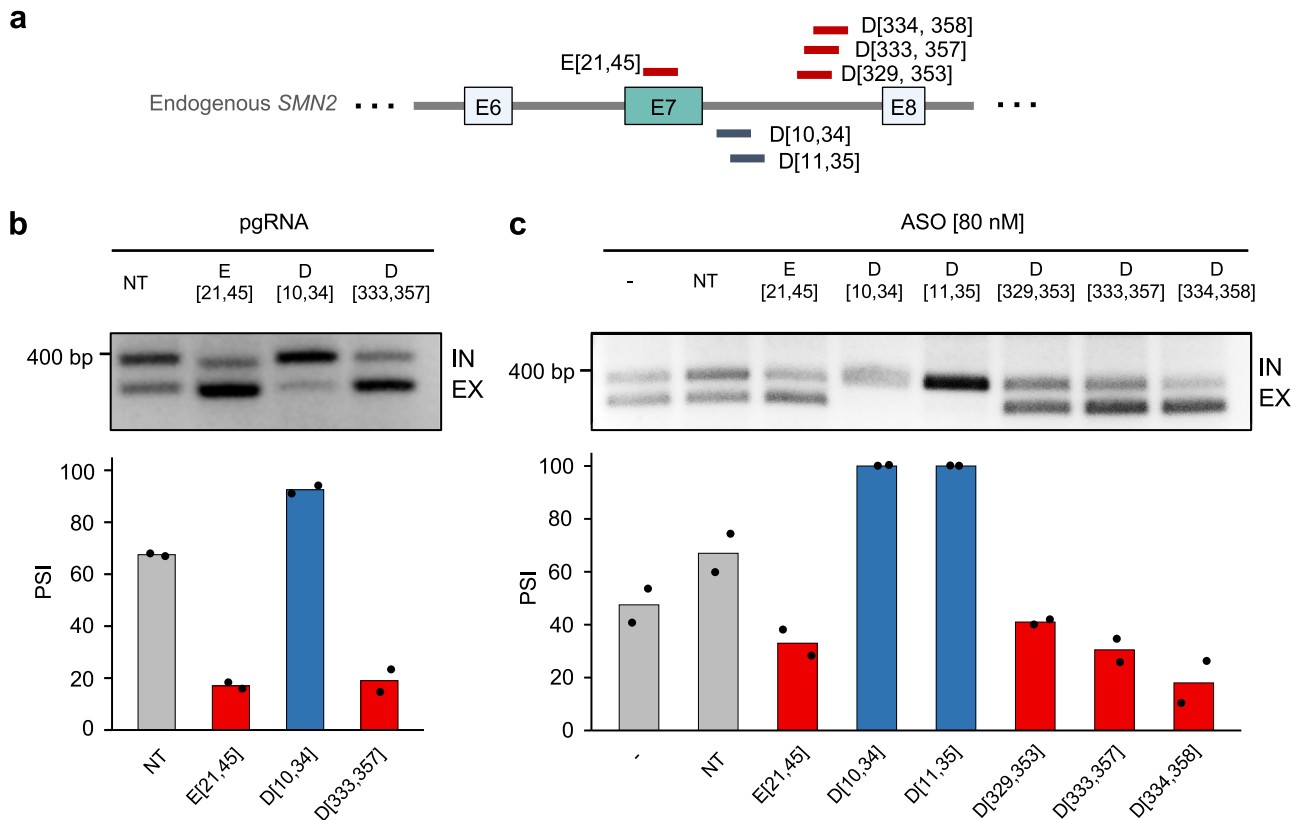

**Fig. 6 | Validation of endogenous *SMN2* exon 7 splicing modulation through the distal intronic SRE ISE-D1 using dCas13d and ASOs. a** Schematics illustrating gRNAs and ASOs targeting the distal downstream intronic SRE, ISE-D1. gRNAs and ASOs targeting the exonic SRE and ISS-N1 region are also included for comparison. **b**, **c** Validation of a gRNA D[333, 357] and three ASOs (D[329, 353], D[333, 357], D[334, 358]) targeting the distal intronic SRE. **b** gRNAs E[21,45], D[10,34], and D[333,357] were individually transfected together with dCas13d into HEK293T cells to target the endogenous *SMN2* pre-mRNA. Representative gel images of RT-PCR

products measuring *SMN2* endogenous isoforms are shown at the top, and quantification of endogenous *SMN2* exon 7 splicing from two replicates of independent transfections is shown at the bottom. The restriction enzyme DdeI, specific for *SMN2*, was used to differentiate *SMN1* and *SMN2* transcripts. **c** A total of six ASOs [80 nM] targeting the *SMN2* region were individually transfected into HEK293T cells. The endogenous *SMN2* exon 7 splicing levels were measured using RT-PCR (top) and quantified in the bar plot using two replicates of independent transfections (bottom).

best of our knowledge, no current approach can systematically map SREs regulating a gene or exon of interest at scale, which is of interest because of the  importance of the native sequence context on the function of an SRE. Exhaustive mapping of SREs for a gene or exon of interest, or an "exon-centric" approach, is not only complementary to the RBP-centric approach in generating functional annotations of the human genome but also has important applications such as identifying regions that can be most effectively targeted by ASOs or other precision medicine therapeutic approaches. The SpliceRUSH platform provides a scalable alternative to the current ASO walk method, enabling us to identify target sequences in the proximal region and distal intronic sequences.

While dCas13-based high-throughput screening tools have much potential, like CRISPRa and CRISPRi technologies using dCas9, several technical challenges were addressed during this study. We encountered limited lentivirus packaging efficiency when using a lentiviral vector containing both dCas13d and gRNA, possibly because the remaining endonucleolytic activity of dCas13d associated with gRNA processing cleaves the viral genome. The limitation of the one-vector system was also recently noted in an independent study[43]. To overcome this issue, we used two separate lentiviral vectors for dCas13d and gRNA expression, which were sequentially transduced into cells, similar to recent Cas13d-based knockdown screens published during the course of this study[41–47]. Another key issue that made dCas13d-based screening more challenging, compared to Cas13d-based knockdown screening, is likely the requirement for more stable binding of dCas13d/gRNA and target sites to compete effectively with cognate RNA-binding effectors. While this can be achieved more easily for individual gRNAs through overexpression of both dCas13d and gRNAs, it is more difficult in the context of a lentiviral-based screen, in which gRNAs must be delivered at a low MOI to ensure that each cell is transduced with one viral particle. We generated stable cell lines with high dCas13d expression to mitigate this issue in our screen. Our analysis also suggests that longer gRNAs of 25-nt provided a much higher signal-to-noise ratio than shorter gRNAs of 22-nt, suggesting that longer sequences help stabilize the gRNA or form a more stable dCas13d/gRNA/target RNA ternary complex. The effect of spacer length on the efficacy of splicing modulation observed at a low MOI with more moderate gRNA expression appears more evident than the differences observed upon overexpression of both dCas13d and individual gRNAs[13,42,44,46,69].

Using *SMN2* exon 7 splicing as a model system, we demonstrate that the SpliceRUSH method produces highly reproducible results, identifying not only previously known SREs in the exon and proximal intronic regions but also previously unknown SREs in distal intronic regions. The effect of gRNAs on modulating splicing is highly sequence-specific, and small shifts in the targeted region can frequently result in drastic differences. Importantly, targeting the identified elements modulates the splicing of both the minigene splicing reporter and the endogenous *SMN2* gene, and similar effects can be achieved using ASOs to target the same region. These results demonstrate that SpliceRUSH can be used as a screening platform for splicing-modulating ASO discovery.

The current screening platform has several limitations concerning false negatives. In addition to the exonic splicing enhancer and ISS-N1, several additional splicing enhancers and silencers with weaker splicing-modulating effects were identified in previous ASO screens[62,63] but missed in our screen. We also noticed that targeting the 5′ splice site resulted in moderate exon skipping, while targeting the 3′ splice site did not show a detectable effect in our screen. This observation is in line with results from our transfection experiments, which showed moderate exon skipping when splice sites were targeted. Similar observations have been made in previous ASO screens for exon skipping therapies, such as *DMD* for Duchenne muscular dystrophy and *SCN1A* for Dravet syndrome[13,16]. A possible explanation of these results

is that dynamic interactions between RNA and RBPs involved in the early stages of splicing regulation can be more readily hindered by gRNAs or ASOs. However, it becomes more challenging to compete with these interactions once stable complexes are formed between spliceosomal proteins and their substrates. Nevertheless, this platform successfully identified SREs with large regulatory effects, which are the most relevant for targeted splicing modulation in translational applications.

While this study focused on the screening of SREs, it is expected that the same strategy may be applicable for the screening of RNA-regulatory elements in the 5′ and 3′ untranslated regions (UTRs) regulating additional steps of RNA processing and gene expression, such as mRNA stability and translation. This exon-centric approach provides a technological platform complementary to the RBP-centric approaches to improve the annotation of RNA-regulatory sequences in the human genome, advance our understanding of RNA-regulatory mechanisms, and facilitate the development of RNA-based precision medicine.

## Methods
### Cloning of Dual-IN and Dual-EX *SMN2* splicing fluorescent reporters
To generate the dual-color splicing reporters, we first prepared a backbone vector, pTomatoSlicer, that expresses both eGFP and tdTomato fluorescent proteins under the control of a CMV early enhancer/chicken ß-actin (CAGGS) promoter. To clone the fluorescent proteins, we synthesized a codon-optimized eGFP sequence fused to an internal ribosome entry site (IRES) and a codon-optimized tdTomato with corresponding overhangs for integration using NEB HiFi DNA assembly (Gibson Assembly Master Mix NEB Cat #E2611). We first integrated the eGFP-IRES into a linearized backbone plasmid, which was transformed into chemically competent *E.coli*, and the intermediary plasmid was then purified. The correct integration was confirmed by Sanger sequencing. TdTomato was subsequently integrated using the same procedure.

Modified *SMN2* sequences were cloned using Gibson assembly into the backbone vector pTomatoSlicer (IRES) through the multiple cloning site (MCS) NheI site. The pCI-*SMN2*-WT plasmid was a gift from Dr. Adrian R. Krainer (CSHL, Cold Spring Harbor, NY). Detailed modifications of the *SMN2* reporter sequences are as follows (Supplementary Fig. 1a; complete sequences can be found in Supplementary Data 1). For the Dual-IN reporter, we inserted 1 nucleotide 'G' at position 37 in exon 7 and 2 nucleotides 'GA' at the beginning of exon 8. The 37G insertion eliminates an in-frame stop codon in the wild-type SMN1/2 exon 7 and also allows exon 7 to become a frame-shifting exon, while the 'GA' insertion in exon 8 keeps the tdTomato sequence downstream of exon 8 in-frame when exon 7 is included. When exon 7 is skipped, the downstream tdTomato sequence is shifted out of the reading frame. There was also a replacement of a portion of exon 6 with a 3x FLAG sequence (region 22–87), given that our initial experiments with reporters containing wild type exon 6 sequence showed protein aggregation when the reporter was transfected at high concentration. For the Dual-EX reporter, the 'GA' nucleotides added at the beginning of exon 8 were removed, while the other Dual-IN modifications were kept. As a result, the tdTomato sequence is in-frame when exon 7 is skipped but out of frame when exon 7 is included. All plasmids were transformed into NEB stable competent *E. coli* (C3040) and isolated using Qiagen plasmid midi kit, according to the manufacturer's instructions. The plasmids were confirmed using Sanger sequencing (Eton Bioscience Inc., Union, NJ).

### Cloning of dCas13d-BFP
The NLS-d*Rfx*Cas13d-eGFP plasmid (Addgene #109050) was modified to replace eGFP with TagBFP fluorescent protein. The eGFP sequence was excised by digesting the vector with NheI and KpnI, and was then

gel extracted. The product was then Gibson assembled with the fragment containing overlaps for 2A, the entire TagBFP sequence, and WPRE overlaps (Gibson Assembly Master Mix NEB Cat #E2611). The assembled product was transformed into NEB Stable Competent *E. coli* (C3040). Plasmids were extracted using Qiagen plasmid midi kit, according to the manufacturer's instructions. The resulting d*Rfx*Cas13d-BFP (dCas13d-BFP) plasmid was confirmed using Sanger sequencing (Eton Bioscience Inc., Union, NJ).

### Cloning of individual gRNAs for transfection
Oligonucleotides corresponding to individual gRNAs were synthesized (IDT) and cloned in the CasRx gRNA cloning backbone (Addgene #109053) using BbsI restriction sites. All constructs were confirmed by Sanger sequencing (Eton Bioscience Inc., Union, NJ). gRNA targeting sequences can be found in Supplementary Data 1.

### Cloning of gRNA lentivirus library
The *Rfx*Cas13d gRNA library targeting the *SMN2* region included in the splicing reporters was designed to cover the entire region (835 bp) with gRNAs of sizes 22-nt and 25-nt tiling with a 1-nt step ($N = 1625$). Non-targeting (NT) controls with sequences that do not align to the human genome (hg19) and sense controls of *SMN2* were included (control total $N = 312$). An Agilent SurePrint oligonucleotide library was synthesized with overlaps for *Rfx*Cas13d Direct Repeat (DR), gRNA sequence targeting *SMN2* minigene (sized 22-nt or 25-nt) or control, and a 3′ overlap with the gRNA backbone (see sequences in Supplementary Data 2). The gRNA oligonucleotide library was amplified using NEB High-Fidelity Q5 polymerase (M04792) with a total of 22 amplification cycles. After the fragments were purified with the QIAquick PCR purification kit (Qiagen 28104), they were Gibson assembled into the pLenti-Rfx-guide-puro backbone digested with BsmBI (Addgene #138150). Library representation in the plasmid pool was confirmed by amplifying the gRNA cassette from the plasmid pool DNA and Illumina sequencing of the PCR product (Genewiz EZ AmpliSeq).

### Lentivirus packaging
For lentivirus production, $2 \times 10^6$ HEK293T cells were seeded per 10-cm plate one day before transfection of the packaging and transfer plasmids. The transfer plasmid (10 μg) and packaging plasmids, pMDL (4.5 μg), pRSV (1.8 μg), and pVSV-G (2.7 μg) were transfected with PEI (packaging amounts listed are for one 10-cm plate). Media was collected at 48 h and 72 h post-transfection and filtered with Millipore Express Plus 0.22 μM pore size (Millipore; GPWP09050), and lentivirus was concentrated with lenti-X concentrator (Takara #631232) to 1/100th of the initial media volume, and the lentivirus was resuspended in PBS. Lentivirus was stored at −80 °C until use.

### Antisense oligonucleotides (ASOs)
ASOs were ordered from IDT. The ASO targeting ISS-N1 shown in Fig. 2b contains 2′-OMe modifications, and all other ASOs shown in Fig. 6b contain 2′-MOE modifications. ASO sequences are listed in Supplementary Data 1.

### Cell culture
Human cells, obtained from ATCC (HEK293T; CRL-3216) and ThermoFisher (Flp-In T-REx 293; R78007), were maintained in Dulbecco's modified Eagle's medium (DMEM) and supplemented with 10% FBS at 37 °C in a 5% $CO_2$ cell-culture incubator.

### Transfections and splicing analysis using RT-PCR and flow cytometry
HEK293T cells maintained in Dulbecco's modified Eagle's medium (DMEM) and supplemented with 10% FBS were seeded one day before transfection (about $1.25 \times 10^5$ cells per well of a 12-well plate). A plasmid ratio of the Dual-IN *SMN2* splicing reporter plasmid:gRNA:dCas13d (25 ng:800 ng:800 ng) was transfected using Lipofectamine 3000 according to the manufacturer's instructions. ASOs were also transfected at final concentrations ranging from 5 to 80 nM using Lipofectamine 3000. After 48 h post-transfection, cells were harvested for further splicing analysis either through RT-PCR or FACS.

For RT-PCR analysis of the splicing products, total RNA isolation was performed using TRIzol reagent (Invitrogen 15596026) and the Direct-zol RNA kit (Zymo Research). cDNA was prepared using SuperScript III reverse transcriptase (Thermo Fisher Scientific 18080044) with random hexamer primers. To measure exon inclusion of the cassette exon of interest (e.g., *SMN2* exon 7), primers to amplify the splicing isoforms of interest were used with Promega Go Taq polymerase (Thermo Fisher Scientific; PRM7123). For RT-PCR analysis of Dual-IN and Dual-EX splicing reporters, we used primers with minigene-specific backbone sequences to avoid amplifying the endogenous *SMN1/2* gene, as listed in Supplementary Data 1. For RT-PCR analysis of the endogenous *SMN2* gene, we used primers complementary to exon 5 and exon 8 sequences (Supplementary Data 1). To distinguish the endogenous *SMN1* and *SMN2* transcripts, the amplified PCR products were digested with DdeI, which specifically digests *SMN2* exon 8. PCR products were resolved on 1.5–2% agarose gel and visualized with ethidium bromide using UVP BioDoc-It imaging system. Exon inclusion levels were quantified with Fiji.

Cells were harvested 48 h post-transfection for flow cytometry to analyze the splicing levels with fluorescence. Cells were washed with PBS, detached with trypsin 0.05% (Corning 25-052-Cl), and resuspended in cold PBS after centrifugation at $500 \times g$ for 3 minutes. Resuspended cells were filtered through a 35 μM cell strainer and stained with either propidium iodide (Sigma-Aldrich P4864) or SYTOX deep red stain (Thermo Scientific, S11380) for dead/alive cells. Cells were sorted on BD FACS Aria III using a 100-micron nozzle. Further downstream processing of flow cytometry data was done using FlowJo.

For flow cytometry analysis, cells were first gated by forward and side scatter to exclude debris and select for single cells. Cells were gated for the PI/SYTOX red dead cell stain to select for live cells. Additionally, cells were selected for high expression of BFP, as that indicated expression of dCas13d. At least 2400 cells were analyzed for each sample. To perform a more quantitative comparison, we estimated the exon inclusion level based on the ratio of tdTomato to eGFP fluorescence on the $\log_2$ scale, i.e., $\log_2(PSI) = \log_2(tdTomato) - \log_2(eGFP) + c$ (c is a constant), as visualized by the MA-contour plots (e.g., Fig. 2d, g, bottom panels). We also calculated the changes in $\log_2(PSI)$ upon dCas13d/gRNA treatment for comparison with the quantifications using RT-PCR (Fig. 2e).

### Generation and validation of DOX-inducible *SMN2* splicing reporter and d*Rfx*Cas13d-BFP stable cell line
**DOX-inducible splicing reporter cell lines.** To generate doxycycline (DOX)-inducible splicing reporter cell lines, the Flp-In T-REx 293 cells were transfected with the Flp-In expression vector containing Dual-IN or Dual-EX *SMN2* splicing reporter along with the Flp recombinase vector, pOG44 (Invitrogen). After selecting positive clones according to Flp-In T-REx manufacturer's instructions, single clones were expanded. Exon 7 splicing was assayed at varying levels of DOX using RT-PCR and FACS to validate the inducible expression of the splicing reporter (Supplementary Fig. 5).

**dRfxCas13d-BFP stable cell line.** The Flp-In T-REx 293 cell line expressing the DOX-inducible splicing reporter was transduced with lentivirus containing d*Rfx*Cas13d-BFP. A titration of lentivirus measured through BFP-positive cells was done to estimate a multiplicity of infection (MOI) of 10 for the dCas13d-BFP monoclonal line. Single BFP-positive clones were sorted into a 96-well plate, and the monoclonal lines were expanded for further validation with fluorescence, flow cytometry, and western blot (Supplementary Fig. 8).

To confirm protein expression via western blot, cells were resuspended in 50 μl lysis buffer (150 mM NaCl, 0.5% Triton X-100, 0.1% sodium dodecyl sulfate (SDS), 50 mM Tris pH 7.5, 1 mM EDTA, 1 mM dithiothreitol (DTT), 1× cOmplete protease inhibitors (Roche Diagnostics, Mannheim, Germany)) for protein extraction. The protein samples were prepared with 4× Laemmli Sample Buffer (Bio-Rad Laboratories, Hercules, CA) and 89 mM B-mercaptoethanol, boiled, and 30 μg of the sample was loaded into 4–12% SDS-PAGE Novex Bis-Tris gels (Invitrogen Cat No. NP0321). After protein transfer onto 0.45 μM nitrocellulose membrane (GE Healthcare), the primary antibody mouse anti-HA (Sigma-Aldrich, H9658, 1:2,000) and mouse anti-alpha-tubulin (Sigma-Aldrich, T6074, 1:10,000) were used. After incubating with goat anti-mouse secondary (Thermo Fisher 31436; 1:10,000 dilution) for 1 h and washing with PBS-Tween 0.05%, Pierce ECL western blotting substrate (Thermo Scientific; 32109) was used to develop with an automatic film processor (Kodak X-OMAT 1000A processor; exposure 5 min).

### gRNA lentivirus transduction

**Individual gRNA lentivirus transduction.** To test whether exon inclusion can be modulated effectively in both positive and negative directions using viral delivery and lower expression of a gRNA, we delivered individual gRNA sequences through lentivirus. Lentiviruses expressing the several individual gRNAs, including non-targeting (NT), the sigRNA E[22,40], and segRNA D[9,33], were packaged individually. To test how the levels of gRNA impact splicing modulation, a low and high gRNA MOI was tested (MOI = 0.1 vs. MOI = 3). After puromycin [1.3 μg/ml] selection, cells were seeded, and media with puromycin [1.3 μg/ml] and DOX [0.5 ng/ml] was added. After 48 h of DOX induction, cells were sorted, and a typical gating scheme for alive singlets expressing BFP was followed. The tdTomato relative to eGFP fluorescence intensity was used to compare splicing-modulating gRNAs (E[22,40] and D[9,33]) and NT gRNA control. RT-PCR was also performed on the cells containing DOX-inducible Dual-IN SMN2, the transduced individual gRNA, and stably expressed dCas13d-BFP.

**Mini gRNA library.** The lentivirus packaging for the mini-gRNA library contained three gRNAs at a ratio of 8:1:1, including 80% NT gRNA (8 μg) and 10% sigRNA E[22, 40] (1 μg), and 10% segRNA D[9,33] (1 μg) of transfer plasmid. The monoclonal Dual-IN SMN2/dCas13d-BFP cell line was seeded and transduced a day later with the mini-gRNA lentivirus at an MOI of 0.3. After selection with puromycin [1.3 μg/mL] 24 h post-transduction, ~30% of surviving cells were passaged under puromycin selection.

Cells expressing the mini-gRNA library, DOX-inducible Dual-IN SMN2 splicing reporter, and stable expression of dRfxCas13d-BFP were then seeded with media containing puromycin [1.3 μg/mL] and DOX [1 ng/mL] to induce splicing reporter expression. After 48 h, cells were sorted to select the subpopulation for BFP expression. Among the BFP-positive cell population, the top 5% and bottom 5% of cells were binned by tdTomato vs. eGFP fluorescence, and the unsorted cells were collected for gRNA sequencing analysis. For each sample, the genomic DNA was extracted using Monarch Genomic DNA purification kit (NEB T3010), followed by PCR amplification of the gRNA cassette using primers compatible with Illumina sequencing and NEB Q5 Hot Start High-Fidelity master mix (NEB M0494) (see primers listed in Supplemental Data 1). After using QIAquick PCR purification for clean-up, samples were submitted to Azenta Life Sciences for Illumina amplicon sequencing (Genewiz Amplicon-EZ). Two independent mini gRNA library transductions were performed for this experiment.

For mini-library screen data analysis, reads with perfect matches to the gRNAs included in the lentivirus pool (i.e., NT, E[22,40], and D[9,33]) were counted and normalized to obtain read per million (RPM) values for each sample for the unsorted, top bin, and bottom bin. For each gRNA, the gRNA enrichment levels were then normalized by dividing the RPM value from the 'top' and 'bottom' bins by the respective RPM value from the 'unsorted' bin to account for differences in the input condition: $\log_2(RPM_{top}/RPM_{unsorted})$ and $\log_2(RPM_{bottom}/RPM_{unsorted})$. Finally, the difference in gRNA enrichment was calculated between the top vs. bottom bins for each gRNA. The average and standard deviation of the mean (SEM) from two replicate screens are shown in Fig. 2h.

**SMN2 gRNA lentivirus library.** The monoclonal cell line with stably expressed dCas13d-BFP, and the inducible dual-fluorescent SMN2 splicing reporter (either Dual-IN or Dual-EX) was transduced with the pooled SMN2 gRNA library lentivirus (n = 1937) at MOI = 0.3. Two independent transductions were done for each reporter cell line (a total of four independent transductions). After 24 h of transducing $20 \times 10^6$ cells per replicate with the SMN2 gRNA library, puromycin (Sigma-Aldrich P8833) was added [1.3 μg/ml] to the media to select for gRNA expressing cells, with an approximate 30% of the cell population surviving, ensuring ≥1000× guide representation in the final surviving cells. Cells were passaged, and the initial cell representation was maintained under puromycin selection for a maximum of two weeks before sorting.

### SpliceRUSH screening and readout

To perform the screening using the dual-fluorescent splicing reporters, cells expressing DOX-inducible Dual-IN or Dual-EX SMN2 reporter, stably integrated dCas13d-BFP, and gRNA-puromycin were sorted using BD FACSAria III. Cells were seeded 48 h before sorting, and at the time of seeding, DOX [1 ng/ml] was added to the media to express the Dual-IN or Dual-EX reporters, as dCas13d and gRNAs are constitutively expressed. Cells were prepared for flow cytometry as described above, with SYTOX red used to stain dead cells. A minimum of $2 \times 10^6$ cells were collected for the unsorted sample, estimated at 1000× coverage for the initial input gRNAs. For the sort, BFP-positive cells were fractionated based on tdTomato/eGFP ratio with the top 5% and bottom 5% populations collected. To ensure ~2800× gRNA coverage of the collected bins, a minimum of 280,000 cells were collected for each bin (Supplementary Data 3). To collect enough cells in the top 5% and bottom 5% bins, about $30 \times 10^6$ cells were seeded with a conservative estimate that 40% would remain BFP positive after expansion. After collection, cells were spun at $500 \times g$ for 5 min and stored at −80 °C until the sequencing library was prepared.

For each screen replicate, the samples processed for gRNA sequencing included unsorted cells and top and bottom bins. The genomic DNA was extracted using a Monarch Genomic DNA purification kit (NEB T3010). Two rounds of amplification for gRNA from the genomic DNA were performed. The first PCR amplified a region (547 bp) containing the gRNA. A minimum of 6 PCR reactions were set up per sample, and 200 ng of genomic DNA was used per reaction using NEB Q5 Hot Start High-Fidelity master mix (NEB M0494) with PCR1 conditions: 98 °C for 30 s, 24× (98 °C for 10 s, 66 °C for 30 s, 72 °C for 45 s), 72 °C for 5 min). The PCR reactions for each sample were pooled and cleaned up with the QIAquick PCR purification kit (Qiagen 28104). Using the pooled PCR1 products, the second PCR reaction per sample was performed. 'PCR2' adds the sample-specific barcode, Illumina sequencing adapters, and stagger sequences to prevent sequencing issues associated with lack of sequence diversity (i.e., region of direct repeat). Again, NEB Q5 Hot Start High-Fidelity master mix was used with PCR2 conditions: 98 °C for 30 s, 16× (98 °C for 10 s, 66 °C for 30 s, 72 °C for 45 s), 72 °C for 5 min). PCR primers can be found in Supplementary Data 1. The products from PCR2 were purified using gel extraction (QIAquick gel extraction 28706). The individual sample libraries were quantified using the KAPA qPCR Illumina library quantification kit (Roche 07960140001). Libraries were pooled in equal ratios and sequenced on a NextSeq 550 using a Mid Output kit v2.5 with 1×150 cycles.

## Bioinformatics analysis of *SMN2* gRNA library screens

For each screen replicate, raw reads from the FASTQ file were demultiplexed based on the Illumina i7 index into different samples, corresponding to the three bins: (1) unsorted, (2) top 5% bin, and (3) bottom 5% bin. The adapters and PCR priming sites were trimmed from the raw reads using *cutadapt 3.4* (https://cutadapt.readthedocs.io). The number of reads perfectly matching each gRNA in the *SMN2* gRNA library was then counted using the script count_spacers.py (Python 3.8.13) described previously[73], and reads per million (RPM) were calculated for each gRNA. The read counts for all replicate screens are summarized in Supplementary Data 4, and additional read statistics are provided in Supplementary Figs. 13 and 17.

To identify gRNAs enriched in the top or bottom bins compared to the unsorted bins, we first calculated an enrichment score for each gRNA. Denoted by the read count for gRNA $g$ in the top, bottom, and unsorted bins $n_{t,g}$, $n_{b,g}$ and $n_{u,g}$, and the total number of reads in the three samples, $N_t$, $N_b$ and $N_u$, respectively.

The comparison of the top vs. the unsorted bins is denoted by $r_{t,g} = n_{t,g}/(N_t + N_u)$ and $r_{t,0} = N_t/(N_t + N_u)$. The enrichment score of gRNA $g$ in the top vs. unsorted bins was estimated below based on a Binomial distribution:

$$e_{t,g} = \left(r_{t,g} - r_{t,0}\right)/s_{t,g},  \tag{1}$$

where $s_{t,g} = \sqrt{r_{t,g}(1 - r_{t,g})/(N_t + N_u)}$ is the estimated standard deviation of $r_{t,g}$.

A Z-score was then calculated by normalizing the enrichment score:

$$z_{t,g} = \left(e_{t,g} - m\right)/\sigma,  \tag{2}$$

where $m$ and $\sigma$ are median and scaled median absolute deviation (MAD) of $e_{t,g}$ (i.e., $\sigma = \text{MAD}(e_{t,g})/0.6745$) across all gRNAs in the library, respectively.

We noticed that gRNAs with insufficient read coverage could give outlier Z-scores. Therefore, Z-scores were filtered to keep only those gRNAs with a read count $\geq 5$ in both compared conditions and RPM $\geq 20$ in at least one of the two compared conditions. Similarly, we estimated $z_{b,g}$, the Z-score for the bottom bin in comparison to the unsorted bin.

A Z-score for comparison of the top vs. bottom bins for gRNA $g$ in a replicate screen was then calculated by combining the two Z-scores for the two sorted bins using the Stouffer's method:

$$z_g = \left(z_{t,g}w_{t,g} - z_{b,g}w_{b,g}\right)/\sqrt{w_{t,g} + w_{b,g}},  \tag{3}$$

where $w_{t,g}$ and $w_{b,g}$ are indicators of whether the gRNA $g$ is quantifiable in the respective bins (1 = quantifiable; 0 = not quantifiable). The Z-scores for individual replicate screens were shown in Figs. 3 and 4, and Supplementary Figs. 14 and 18.

Lastly, a final Z-score for gRNA $g$ was calculated by combining the eight Z-scores from the four replicates as follows:

$$Z_g = \frac{z_{t,g}^{IN1}w_{t,g}^{IN1} - z_{b,g}^{IN1}w_{b,g}^{IN1} + z_{t,g}^{IN2}w_{t,g}^{IN2} - z_{b,g}^{IN2}w_{b,g}^{IN2} - z_{t,g}^{EX1}w_{t,g}^{EX1} + z_{t,g}^{EX1}w_{b,g}^{EX1} - z_{t,g}^{EX2}w_{t,g}^{EX2} + z_{b,g}^{EX2}w_{b,g}^{EX2}}{\sqrt{W}},  \tag{4}$$

where superscript *IN1*, *IN2*, *EX1*, and *EX2* are indicators of the four replicates and

$$W = w_{t,g}^{IN1} + w_{b,g}^{IN1} + w_{t,g}^{IN2} + w_{b,g}^{IN2} + w_{t,g}^{EX1} + w_{b,g}^{EX1} + w_{t,g}^{EX2} + w_{b,g}^{EX2}  \tag{5}$$

is the number of bins with quantifiable Z-scores.

The final Z-scores were also filtered to keep only gRNAs with $W \geq 4$ (Fig. 5 and Supplementary Fig. 19). These Z-scores were converted into p-values based on the normal distribution (two-sided), followed by the calculation of false discovery rate (FDR) using the Benjamini–Hochberg procedure[74].

## RNA-seq library preparation and analysis

HEK 293T cells were seeded at about 125,000 cells one day before transfection in a 12-well plate. Plasmids for dCas13d-BFP and gRNA were transfected using Lipofectamine 3000; dCas13d-BFP (800 ng) and individual gRNA (800 ng) were added to each well. The individual gRNA plasmid was either a control non-targeting gRNA or D[333,357] gRNA targeting the downstream intronic region identified in the screen. Stranded poly-A RNA-seq libraries were constructed, with two biological replicates per group, using the Illumina TruSeq Stranded mRNA Library Preparation Kit at Columbia Genome Center. Each library was sequenced on NovaSeq 6000 to obtain ~40 million $2 \times 100$ bp paired-end reads. The resulting RNA-seq reads were mapped with OLego (v1.1.5) using the stranded mode[75]. Gene expression and alternative splicing were quantified with uniquely mapped reads using the Quantas pipeline (v1.1.1) as previously described[76]. For quantification of the *SMN2* exon 7 splicing, we kept multi-mapped reads due to the high sequence similarity of the *SMN1* and *SMN2* genes. The global expression/cassette splicing patterns of samples were then assessed by principal component analysis (PCA) using the 'prcomp' function in R. For this analysis, lowly expressed genes (the maximum of the average group RPKM values below the median) and cassette splicing events with low read coverage (< 20 reads) were excluded. In addition, log transformation was applied to the gene expression values before PCA.

## Reporting summary

Further information on research design is available in the Nature Portfolio Reporting Summary linked to this article.

## Data availability

The data supporting this study's findings are available from the corresponding authors upon request. Illumina sequencing data from the *SMN2* splicing screens and RNA-sequencing have been deposited to the NCBI Short Read Archive (SRA) under accession PRJNA1052769. The reference human genome (hg19) was used for RNA-seq read mapping. Source data are provided in this paper.

## Code availability

The scripts used for this study are available at https://github.com/chaolinzhanglab/rush.

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

## Acknowledgements

We thank Patrick Hsu for sharing dCas13d/gRNA expression plasmids and Adrian R. Krainer for the pCI-SMN1/2 minigene plasmids. We also thank Peter Sims, Alejandro Chavez, and Samuel H. Sternberg for helpful discussion and advice throughout the project. This study was supported by grants from the National Institutes of Health (NIH) (R56HG012359, R01HG012359, and R35GM145279 to C.Z.) and a pilot grant from the Columbia Precision Medicine Initiative (C.Z.). Y.R. was supported by the National Science Foundation (NSF) Graduate Research Fellowship Program (GRFP). This research used the Columbia Genomics and High Throughput Screening Shared Resource and CCTI Flow Cytometry Core, was supported in part by NIH awards S10RR027050 and S10OD020056, and P30CA01369.

## Author contributions

Conceptualization and experimental design: Y.R., D.U., C.Z. Cloning of splicing reporters: Y.R., D.U., X.W., M.J. and L.V.Y. Experimental work related to dCas13d/gRNA, screening and validation: Y.R., D.U. and Y.T.Y. Analysis: Y.R., Y.T.Y., Q.W. and C.Z. Writing: Y.R. and C.Z. All authors critically reviewed the manuscript.

## Competing interests

Y.R., D.U., and C.Z. are inventors on a patent application submitted based on this work. C.Z. is a co-founder of DAYI Therapeutics, Inc. The other authors declare no competing interests.
