## [Peer Review File · Nature Communications]

CRISPR-dCas13d based deep screening of proximal and distal splicing-regulatory elementsREVIEWER COMMENTS

Reviewer #1 (Remarks to the Author):

Pre-mRNA splicing modulation via ASO and drugs is an effective method for disease therapy. However, the identification of splicing-regulatory elements (SREs) in native sequence context are challenging. In this manuscript, the authors developed an CRISPR-RfxCas13d based SREs high-throughput screening method called SpliceRUSH with the principle of competing with endogenous RBP through steric hindrance. By applying this method to SMN2 SREs screening, they not only identified the known SREs, but also discovered a novel distal intronic splicing enhancer, D[332,356]. Overall, although the concept of CRISPR-RfxCas13d based screening is not new. For example, has been developed for circRNA screening (Li et.al, Nature Method, 2021). But its application for SREs screening is still impressive and may contribute for RNA-based drug discovery. However, the study so far is a bit preliminary, many conditions and optimization should be tested. In addition, the advantage for this screening method compared to ASO walk needs to be highlighted. Detailed comments are shown below:

1. In Figure 1C, the definition of segRNA and sigRNA needs to be explained well. For example, according to the context, splicing-enhancing gRNA means enhance the inclusion of exon 7 instead of enhance splicing.
2. As showed by the author, the length of gRNA will affect the exon 7 inclusion, the authors should test different length of gRNAs instead of only use 22nt and 25nt. This may help improve the screening efficiency of the system.
3. The reason why longer gRNA has better efficiency needs to be determine, does it has better Cas13 protein targeting efficiency or better steric hindrance towards splicing regulators? Or some other reasons?
4. When comparing the Dual-IN and Dual-EX method for screening, the former one has the better Z-score, what's the reason for that? Particularly, D[10,34] is lost in 22nt EX screen (Fig. 18). Is IN has higher working efficiency or higher background.
5. When comparing the exon 7 inclusion percentage using RT-PCR, for example, in fig. S11, the global expression level of IN+EX differs in different conditions, which makes the comparison suspicious. Moreover, whether using ASO or Cas13 method to regulate splicing also will affect the expression of the target gene?
6. In fig.S9 and S11, it looks like low MOI of lenti-gRNA towards D[9,33] does not work well?
7. The comparison of CRISPR-RfxCas13d based screening with ASO walk needs to be performed. The Cas13d method includes more components in the system, does it has higher screening efficiency? Which one is easier to perform?

Reviewer #2 (Remarks to the Author):

In this manuscript, Recions et al. presented SpliceRUSH, a dCas13d-based screening method to systematically map splicing regulatory elements of SMN2 exon 7. Although a similar concept for mapping gene-centric functional elements or domains has been demonstrated with Cas9-based DNA-targeting tilling scans, applications for exon-centric splicing elements have not been demonstrated. The authors used dual color splicing reporters of SMN2 exon 7 and screened a library containing >800 on-target gRNAs. Remarkably, two different versions of splicing reports with opposite readouts of splicing changes obtained consistent hits. Altering spacer length from 22nt to 25-nt identified more hits and most were agreeable. Interestingly, the authors found a new deep intronic splicing element from the screen and showed indeed blocking this element reduced exon 7 splicing. Such deep intronic elements would be difficult to reveal without tilling screens using a method like SpliceRUSH. Overall, I think this is an elegant study that makes a clever use of CRISPR tilling screen to identify splicing elements. The manuscript is well written. I have a few comments to strengthen the manuscript.

Major

The goal of such screens is to identify the most effective targeting sequences. Naturally, readers will want to know how the new ASOs the authors identified are compared to the already approved nusinersen? Because HEK293T cells can't answer this question due to the already high basal inclusion, this may need to be tested in a more relevant system that the SMA field has established.

The main discovery of the manuscript is the deep intronic element. Can the authors provide more evidence (e.g., deletion in minigenes or endogenous locus) beyond simple steric hindrance to exclude the possible off-target effect of gRNA/ASO, and also provide insight about the cognate trans-factors?
Minor

In Fig 2e, can the authors explain how the FACS PSI value of a treatment group is derived from the cell population?

In Fig2h, can the authors explain how the gRNA enrichment was normalized and derived? were they compared to NT distribution in respective bins? If not, can the authors make this comparison?

In Fig2d, figs3a, figs9a it is difficult to distinguish gRNA treated cells from NT cells in the figure. A traditional way presenting data of this sort is separating density plots for each group.

Can the authors explain more about what they mean "in a native sequence context" in the title when the screen is based on a reporter minigene?

Reviewer #3 (Remarks to the Author):

In this work, Recinos et al. developed a platform (the so-called SpliceRUSH in the manuscript) for high-throughput screening splicing regulatory elements (SREs) in a specific pre-mRNA of interest using the dCas13d/gRNA system. Systematical identification of SREs using this method will help us to understand the mechanisms and functions of splicing regulation. Besides, applying this method as a pre-selection procedure could save time and costs for discovering effective splice-switching antisense oligonucleotides (ASOs) through significantly reducing the number of ASO candidates to a testable size. The authors applied this method to SMN2 gene, the disease gene of SMA, whose splicing regulation has been extensively studied. After optimizing and establishing this method, they identified one known proximal splicing enhancer in exon 6, one known proximal splicing silencer in intron 7 (where Sprinraza ASO targets), and one novel distal splicing silencer in intron 7. Overall, the platform was carefully designed and optimized, and the screening was successfully performed for SMN2 gene. I have several concerns about the screening results of SMN2 gene and its application potential on other genes.

Specific points

1. The authors performed two SREs screens using Dual-IN and Dual-EX reporters, respectively. Each screen has two replicates. Although the scale bars of Z-scores from 4 individual replicates are not the same (Fig 3b and 4c), one could see that there are gRNA hits with Z-scores comparable to or even higher than segRNA D[10, 34] hit. The targeting positions of several gRNA hits look conserved among 4 replicates. Have the authors validated the effects of these gRNAs on exon 7 splicing? Are they true hits?

2. Why the enrichment of segRNAs is so moderate in 4 screens?

3. The authors should provide more detailed information about the previously identified SREs in SMN2 gene for comparison. For example, previous studies from Krainer's lab identified several splicing silencers in exon 7, intron 6, and intron 7 using ASO tiling method (Ref. 64, 65). It seems that the spliceRUSH method only identified the SREs with the strongest regulatory effects in the case of SMN2. Is this method not sensitive enough to capture the SREs with moderate effects? This issue is important for future studies on understanding of splicing regulation using the method.

Other points

1. In the Introduction, several descriptions are not precise.

-Page 3, line 57: RNA splicing is required not only for protein translation but also for the maturation of

many lncRNAs and etc.

-Page 3, line 61-63: The authors did not fully describe the roles of SREs in splicing regulation. Besides recognized by splicing regulators, they have been shown to modulate splicing by forming secondary structures even through long range interactions.

2. The quality of most agarose gel images is low in the manuscript. The bands are not sharp enough, and the background of several images is relatively high (e.g. Fig. 6c). To determine splicing changes quantitatively, RT-PCR condition need to be optimized to avoid over-amplification and PCR products need to be clearly separated by agarose gel electrophoresis. Otherwise, the resulting PSI values might not be accurate for comparison.

3. Fig 1a: RBPs can either enhance or inhibit the inclusion of alternative exons through binding to SREs. The arrow head in the first cartoon usually refers to "inhibitory effect", which may cause misleading.

4. Fig. 2g: The dark green color is hardly visible. What do the two gray rectangles indicate in the right FACS plot?

5. Page 28, line 908: D[22,40] and E[9,30] are typos.

Point-by-point responses to Reviewers' comments

Reviewer #1:

Reviewer #1 (Remarks to the Author):

Pre-mRNA splicing modulation via ASO and drugs is an effective method for disease therapy. However, the identification of splicing-regulatory elements (SREs) in native sequence context are challenging. In this manuscript, the authors developed an CRISPR-RfxCas13d based SREs high-throughput screening method called SpliceRUSH with the principle of competing with endogenous RBP through steric hindrance. By applying this method to SMN2 SREs screening, they not only identified the known SREs, but also discovered a novel distal intronic splicing enhancer, D[332,356]. Overall, although the concept of CRISPR-RfxCas13d based screening is not new. For example, has been developed for circRNA screening (Li et.al, Nature Method, 2021). But its application for SREs screening is still impressive and may contribute for RNA-based drug discovery. However, the study so far is a bit preliminary, many conditions and optimization should be tested. In addition, the advantage for this screening method compared to ASO walk needs to be highlighted. Detailed comments are shown below:

We thank the reviewer for their very positive feedback of our study and pointing out previous CRISPR/Cas13d-based high-throughput screenings. We would like to highlight that all relevant previous efforts used nuclease active Cas13d for gene knockdown, while SpliceRUSH is the first pooled high-throughput screening using the nuclease inactive effector. These two approaches have fundamentally different requirements, e.g., binding affinity between Cas13d (or dCas13d)/gRNA and the target RNA. As one would expect, transient interaction might be sufficient for gene knockdown, but efficient splicing modulation by steric hindrance would require more stable dCas13d/gRNA interacting with the target and blocking the respective splicing-regulatory elements. In fact, this was the area we made substantial efforts to optimize the assay and ensure the proper stoichiometry of dCas13d/gRNA and the splicing reporter by developing multiple stable cell lines. These reagents are instrumental to achieve the success we reported in the manuscript.

We believe our results have provided compelling evidence that CRISPR/dCas13d can specifically and robustly modulate splicing in a format that is compatible with high-throughput screening. On the other hand, we completely agree with the reviewer that additional biological and technical improvements would further enhance the performance of this system, and we are continuing working hard in this direction for follow-up studies. However, considering that the CRISPR technologies evolve very rapidly and the importance of the topic, we believe publication of our results as the first proof-of-principle of CRISPR/dCas13d for lentivirus-based pooled high-throughput screening would be very informative and make widespread impact on the research community by encouraging other groups to join our efforts. This would speed up further improvement of the technology and development of additional biological applications.

1. In Figure 1C, the definition of segRNA and sigRNA needs to be explained well. For example, according to the context, splicing-enhancing gRNA means enhance the inclusion of exon 7 instead of enhance splicing.

We define splicing-enhancing or splicing-inhibiting gRNA (segRNA and sigRNA) in the similar sense as compared to other terminologies widely used in the field, such as exonic splicing enhancers or silencers (ESE and ESS). Here ‘splicing enhancing’ refers to enhance inclusion (i.e., splicing in) of the alternative exon. In response to the reviewer’s comment, we have revised the descriptions to convey that the intended segRNA/sigRNA function is specific with regard to the alternative exon 7 to avoid potential confusions.

“With the Dual-IN reporter, we expect that splicing-enhancing gRNAs (segRNAs) that increase exon 7 inclusion will be enriched in the top bin with a high tdTomato/eGFP ratio, whereas splicing-inhibiting gRNAs (sigRNAs) that decrease exon 7 inclusion will be enriched in the bottom bin with low tdTomato/eGFP ratio.”

2. As showed by the author, the length of gRNA will affect the exon 7 inclusion, the authors should test different length of gRNAs instead of only use 22nt and 25nt. This may help improve the screening efficiency of the system.

It has been established in the recent literature that the gRNA length can significantly influence their targeting efficiency (e.g., (Zhang et al., 2018; Wessels et al., 2020; Li et al., 2021; Zhang et al., 2021)). Noteworthy examples include ref. (Wessels et al., 2020), which individually tested 15 to 36 bp and ultimately screened with 23 bp, and ref. (Zhang et al., 2021), which tested 21 to 30 bp and screened with 24 bp.

Our dCas13d/gRNA screening of *SMN2* exon 7 is one of the few to include two sizes in the same screening experiment, providing a direct and comprehensive comparison of both sizes (Fig. 5b and Fig. 5c). In addition to targeting efficiency, important considerations for our screen also include linking our findings to prospective ASO targets, which are often characterized by their smaller sizes to facilitate their synthesis and delivery. For example, *SMN* ASO tiling screens tested ASO sizes 15-mer to 18-mer (Hua et al. *Am J Hum Genet* 2008).

Lastly, there are adverse effects associated with longer gRNA lengths. This phenomenon has been observed for Cas9 and was also observed in the single-cell screening using Cas13d, as the capture efficiency is tied to capturing the tag located in the 3’ region of the gRNA (Wessels et al. *Nat Methods* 2023). An exonuclease likely interacts with the unprotected portion of the gRNA.

We agree with the reviewer that technology can benefit from additional optimization, as we responded above. For example, in addition to gRNA size, one potential avenue for further improvements in CRISPR-Cas13 screening efficiency is increasing gRNA stability (Mendez-Mancilla et al., 2022; Nelson et al., 2022), one area we are exploring for follow-up studies.

3. The reason why longer gRNA has better efficiency needs to be determine, does it has better Cas13 protein targeting efficiency or better steric hindrance towards splicing regulators? Or some other reasons?

As pointed out in the response above, a longer gRNA sequence, to a certain degree, has been shown to improve targeting efficiency resulting in more effective gene knockdown. Here targeting efficiency refers to both how fast Cas13d/gRNA find its target transcript and bind the

target site with sufficient affinity for its function. Our analysis demonstrated that longer gRNAs of 25-nt provided a substantially higher signal-to-noise ratio compared to shorter gRNAs of 22-nt in the context of steric hindrance and splicing modulation, consistent with previous results from gene knockdown experiments. It is particularly worth noting that the effects of 25nt gRNAs are most correlated with 22nt gRNAs with the same 5' end (Fig. 5b-d and Supplementary Fig. 19). This observation is most compatible with the notion that a longer gRNA facilitates and/or stabilizes the interaction between dCas13d/gRNA and target (and thereby enhancing the efficiency steric hindrance), without altering the register of the dCas13d on gRNA/target RNA. Therefore, we believe the improvement in targeting efficiency and steric hindrance are intrinsically coupled and difficult to distinguish. For our application, the combined effects on splicing modulation efficiency are the most relevant.

4. When comparing the Dual-IN and Dual-EX method for screening, the former one has the better Z-score, what's the reason for that? Particularly, D[10,34] is lost in 22nt EX screen (Fig. 18). Is IN has higher working efficiency or higher background.

The reviewer raised a very intriguing question for which we do not have a conclusive answer. We speculate that both Dual-IN and Dual-EX splicing reporters have relatively high tdTomato intensity at the baseline exon 7 inclusion/exclusion level, and the signal might be saturated when the exon inclusion level further increases (for Dual-IN) or decreases (for Dual-EX). A possible reason of this is, e.g., the relatively long half-life of tdTomato. This explanation is consistent with the better Z-score from the Dual-IN reporter than the Dual-EX reporter, as the range of Z-scores is largely driven by sigRNAs with reduction of tdTomato for the Dual-IN reporter and increase of tdTomato for the Dual-EX reporter, respectively. On the other hand, the detection of gRNAs targeting the ISS-N1 element did appear to have a better signal-to-noise ratio for the Dual-EX reporter (reduction in tdTomato compared to the baseline) than the Dual-IN reporter (increase in tdTomato). Regarding the 22-nt gRNA screens, they all have dramatically reduced sensitivity, resulting in the difficulty in detecting the regulatory effect of D[10, 34] gRNA targeting ISS-N1. This does not seem to be specific for the Dual-EX screens. See also our response to point #2 of reviewer #3.

5. When comparing the exon 7 inclusion percentage using RT-PCR, for example, in fig. S11, the global expression level of IN+EX differs in different conditions, which makes the comparison suspicious. Moreover, whether using ASO or Cas13 method to regulate splicing also will affect the expression of the target gene?

Our primary focus in Supplementary Fig. 11 is to test whether efficiency in splicing modulation, as quantified by the ratio of the inclusion band to the sum of the inclusion and skipping bands, varies based on the level of dCas13d-BFP expression. As the reviewer pointed out, we did observe differences in the total band intensity of the inclusion and skipping forms, which we believe is most likely due to technical variations related to the RT-PCR method (e.g., inaccuracy in quantifying RNA/cDNA concentration to determine the amount of input). Such variations are not uncommon.

To help to address this issue, we have now performed RNA-sequencing to evaluate the impact of dCas13d/gRNA on overall gene expression level. We found that the expression of the target gene remains unaffected, despite the expected change in splicing, upon treatment of dCas13d/gRNA (D[333,357]). We also confirmed that dCas13d/gRNA treatment did not affect global gene expression or splicing. These results are now included in Supplementary Fig. 20 (and described in the text in pp. 10).

6. In fig.S9 and S11, it looks like low MOI of lenti-gRNA towards D[9,33] does not work well?

The influence of gRNA expression through lentivirus transduced at varying MOIs on the effectiveness of splicing modulation is highlighted in Supplementary Fig. 9. The gRNA level is a limiting step in the assay when there is one copy per cell, as was observed by the high vs. low gRNA MOI splicing comparison using D[9,33]. Furthermore, as we identified with our screen, gRNA D[10,34] is more effective at modulating splicing and increasing exon 7 inclusion at low gRNA MOI conditions. The splicing modulation disparity observed at varying lentivirus MOIs underscores the significance of optimizing gRNA expression and/or stability to achieve consistent and robust splicing modulation, which we will continue to investigate for follow-up studies.

7. The comparison of CRISPR-RfxCas13d based screening with ASO walk needs to be performed. The Cas13d method includes more components in the system, does it has higher screening efficiency? Which one is easier to perform?

A comparison of ASO data to gRNA data, as showcased in Fig. 2, illustrates the overall similarities of the two approaches for splicing modulation. This is consistent with the notion that the two approaches have similar modes of action (i.e., steric hindrance by competing with endogenous splicing factors for target RNA binding).

Performing an ASO screening by “ASO walk” is relatively straightforward regarding technology and the required experimental steps. However, ASO screens are notably expensive, averaging around \$250 per modified ASO of approximately 20 bp, and low throughput, as ASOs are individually transfected into cells and evaluated using RT-PCR to quantify their effects on splicing modulation.

The proposed CRISPR-RfxCas13d-based screening overcomes the limitations above because it enables a large library of thousands of gRNAs to be screened as a pool in a single experiment. We would like to highlight that in the current study, we used a dual-color fluorescent reporter as readout, which limited our screen to one exon (but a large number of gRNAs) per experiment. Employing a different readout, such as single-cell sequencing, could significantly increase the throughput of exons tested in a single experiment and also splicing detection sensitivity, offering an avenue for further enhanced efficiency in the screening process.

Reviewer #2 (Remarks to the Author):

In this manuscript, Recions et al. presented SpliceRUSH, a dCas13d-based screening method to systematically map splicing regulatory elements of SMN2 exon 7. Although a similar concept for mapping gene-centric functional elements or domains has been demonstrated with Cas9-based DNA-targeting tiling scans, applications for exon-centric splicing elements have not been demonstrated. The authors used dual color splicing reporters of SMN2 exon 7 and screened a library containing >800 on-target gRNAs. Remarkably, two different versions of splicing reports with opposite readouts of splicing changes obtained consistent hits. Altering spacer length from 22nt to 25-nt identified more hits and most were agreeable. Interestingly, the authors found a new deep intronic splicing element from the screen and showed indeed blocking this element reduced exon 7 splicing. Such deep intronic elements would be difficult to reveal without tiling screens using a method like SpliceRUSH. Overall, I think this is an elegant study that makes a clever use of CRISPR tiling screen to identify splicing elements. The manuscript is well written. I have a few comments to strengthen the manuscript.

We are grateful for the reviewer's very positive feedback and constructed comments.

Major

The goal of such screens is to identify the most effective targeting sequences. Naturally, readers will want to know how the new ASOs the authors identified are compared to the already approved nusinersen? Because HEK293T cells can't answer this question due to the already high basal inclusion, this may need to be tested in a more relevant system that the SMA field has established.

Many of the previous ASO screenings on SMN2 were performed in 293T cells (Hua et al. Am J Hum Genet 2008). Therefore, we used 293T cells to develop our screening platform to provide a proof of concept, leveraging the extensive published data for direct comparison with our results. We would like to highlight that our goal is to develop a general high-throughput screening platform to discovery both proximal and distal SREs that can provide candidates to be targeted by different approaches such as ASOs, rather than improve the current ASO therapeutics for SMA (especially given that there are already three FDA approved drugs for SMA). Our comparison with the previous ASO data suggests that SpliceRUSH successfully identified the previously known SREs, including the ISS-N1 element targeted by nusinersen/Spinraza, without requiring any prior knowledge, as well as a novel element ISE-D1 in the downstream intron. The discovery of the novel element is intriguing, but it is a splicing enhancer rather than a silencer. Therefore, blocking this element would increase exon skipping, so it is not directly suitable for developing SMA therapeutics. Nevertheless, this example provides compelling evidence that this platform can be used to make novel discoveries.

-The main discovery of the manuscript is the deep intronic element. Can the authors provide more evidence (e.g., deletion in minigenes or endogenous locus) beyond simple steric hindrance to exclude the possible off-target effect of gRNA/ASO, and also provide insight about the cognate trans-factors?

We thank the reviewer for this question. We think it is unlikely the discovered intronic element is a consequence of indirect effect, because 1) all other ASOs targeting known SREs resulted in

specific splicing changes in the expected direction; 2) we have achieved consistent and robust splicing modulation using multiple ASOs and gRNAs with different sequences targeting the same region; 3) we have now performed RNA-seq of samples upon ASO treatment in comparison to controls. While we validated the specific splicing change of *SMN2* exon 7, we did not observe global gene expression or splicing changes due to off target effects (Supplementary Fig. 20 in the revised manuscript).

However, in response to the reviewer's request, we performed deletion experiments by removing either the region targeted by the gRNAs/ASOs that reduce exon inclusion (nucleotides 321-365), or a series of 5-nt deletions within the targeted region (Fig. R1a; the deleted regions are highlighted in red). However, the interpretation of the result is not straightforward. We found the full deletion 321-365 did not cause any substantial impact on exon inclusion, while several individual smaller deletions resulted in increased exon inclusion or skipping (Fig. R1b). The disparity of splicing changes resulting from deletions and dCas13d/gRNA has been reported previously (e.g., (Leclair et al., 2020)). In this particular case, this complexity may be attributed to the intricate RNA-secondary structures formed by the downstream intronic sequences, including the targeted region, as demonstrated in a previous study (Fig. R1a; (Singh et al., 2013)). This is also consistent with our attempt to bioinformatically search for candidate RBPs binding to the targeted region using published CLIP datasets (e.g., eCLIP data from the ENCODE consortium), which did not suggest any apparent candidates for cognate trans-factors. These data suggest a more intricate relationship between the element and splicing regulation and further investigations are required to delineate the underlying mechanism, which is beyond the scope of the current study. Due to the inconclusive nature of the results, we decided to include them here for reviewers' information.

Fig. R1: The impact of nucleotide deletions in ISE-D1 on exon 7 inclusion. **a**, Schematic (adapted from ref. (Singh et al., 2013)) showing the predicted RNA-secondary structures formed by the downstream intron 7 sequences. The region targeted by ASOs/gRNAs subject to deletion analysis is highlighted in red, with the nucleotide sequence and coordinate shown in the inset. **b**, RT-PCR analysis of exon 7 inclusion in control and deletion mutants. The full deletion encompasses nucleotides 321-365 (highlighted in red). For individual 5-nt deletions, the position of the first deleted nucleotide is provided for each construct. The gel images from two biological replicates from independent transfections are shown with the quantification of exon inclusion shown at the bottom.

Minor

In Fig 2e, can the authors explain how the FACS PSI value of a treatment group is derived from the cell population?

In Fig 2e, the FACS PSI value of a treatment group was determined based on a more quantitative approach. We calculated the exon inclusion level by estimating the ratio of tdTomato to eGFP

fluorescence on a \log_2 scale, denoted as $\log_2(\text{PSI})$. This calculation involved the equation $\log_2(\text{PSI}) = \log_2(\text{tdTomato}) - \log_2(\text{eGFP}) + c$, where 'c' represents a constant factor. This methodology was depicted and visualized through an MA-contour plot, allowing us to derive the PSI value from the cell population. We further calculated changes of $\log_2(\text{PSI})$ upon gRNA treatment in comparison to control, in which the constant 'c' was eliminated, so that the impact of each gRNA as quantified from FACS sorting can be directly compared to the quantification by RT-PCR, as shown in Fig. 2e. The relevant descriptions in Method (pp. 22-23) have now been revised to improve clarity:

“To perform a more quantitative comparison, we estimated the exon inclusion level based on the ratio of tdTomato to eGFP fluorescence on the \log_2 scale, i.e., $\log_2(\text{PSI}) = \log_2(\text{tdTomato}) - \log_2(\text{eGFP}) + c$ (c is a constant), as visualized by the MA-contour plots (e.g., Fig. 2d,g, bottom panels). We also calculated the changes in $\log_2(\text{PSI})$ upon dCas13d/gRNA treatment for comparison with the quantifications using RT-PCR (Fig. 2e).”

In Fig2h, can the authors explain how the gRNA enrichment was normalized and derived? were they compared to NT distribution in respective bins? If not, can the authors make this comparison?

We apologize for the confusion. Additional details of data analysis are now provided in Method (pp. 24).

“For mini-library screen data analysis, reads with perfect matches to the gRNAs included in the lentivirus pool (i.e., NT, E[22,40], and D[9,33]) were counted and normalized to obtain read per million (RPM) values for each sample for the unsorted, top bin, and bottom bin). For each gRNA, the gRNA enrichment levels were then normalized by dividing the RPM value from the ‘top’ and ‘bottom’ bin by the respective RPM value from the ‘unsorted’ bin to account for differences in the input condition: $\log_2(\text{RPM}_{\text{top}}/\text{RPM}_{\text{unsorted}})$ and $\log_2(\text{RPM}_{\text{bottom}}/\text{RPM}_{\text{unsorted}})$. Finally, the difference in gRNA enrichment was calculated between the top vs. bottom bins for each gRNA. The average and standard deviation of the mean (SEM) from two replicate screens are shown in Fig. 2h.”

In Fig2d, figs3a, figs9a it is difficult to distinguish gRNA treated cells from NT cells in the figure. A traditional way presenting data of this sort is separating density plots for each group.

We agree that the difference between the two cell populations is sometimes difficult to discern visually. For this reason, we also provided the contour density plot after MA-transformation below each FACS scatter plot. Note that the MA-plot simply rotates the original FACS plot for 45 degrees.

Can the authors explain more about what they mean “in a native sequence context” in the title when the screen is based on a reporter minigene?

The 'native sequence context' aims to convey that while the study is conducted within a minigene framework, the context mirrors, to the extent possible, the natural genomic environment where these splicing-regulatory elements exert their regulatory functions. This is to distinguish from the previous studies of SRE or genetic variant screens, which inserted a randomized sequence library or the targeted exon (together with a small extension to the flanking intronic regions) into a heterologous minigene context. In those cases, SREs identified in heterologous context might not work the same as expected from the screening results due to context-dependent regulatory effects. Our reporters include the neighboring exons (exon 6 and exon 8) and the entirety of the intronic sequences (except for a truncation in the upstream intron for the cloning purpose). This strategy can thus maximize the chance of translating our findings to regulation of the endogenous gene. That said, we admit that our current screen does not target the endogenous gene directly due to the requirement of a fluorescent reporter for readout, which we aim to overcome in a follow-up study.

Reviewer #3 (Remarks to the Author):

In this work, Recinos et al. developed a platform (the so-called SpliceRUSH in the manuscript) for high-throughput screening splicing regulatory elements (SREs) in a specific pre-mRNA of interest using the dCas13d/gRNA system. Systematical identification of SREs using this method will help us to understand the mechanisms and functions of splicing regulation. Besides, applying this method as a pre-selection procedure could save time and costs for discovering effective splice-switching antisense oligonucleotides (ASOs) through significantly reducing the number of ASO candidates to a testable size. The authors applied this method to SMN2 gene, the disease gene of SMA, whose splicing regulation has been extensively studied. After optimizing and establishing this method, they identified one known proximal splicing enhancer in exon 6, one known proximal splicing silencer in intron 7 (where Sprinraza ASO targets), and one novel distal splicing silencer in intron 7. Overall, the platform was carefully designed and optimized, and the screening was successfully performed for SMN2 gene. I have several concerns about the screening results of SMN2 gene and its application potential on other genes.

We thank this reviewer for their positive feedback and all the constructive comments, which we have addressed carefully below.

Specific points

1. The authors performed two SREs screens using Dual-IN and Dual-EX reporters, respectively. Each screen has two replicates. Although the scale bars of Z-scores from 4 individual replicates are not the same (Fig 3b and 4c), one could see that there are gRNA hits with Z-scores comparable to or even higher than segRNA D[10, 34] hit. The targeting positions of several gRNA hits look conserved among 4 replicates. Have the authors validated the effects of these gRNAs on exon 7 splicing? Are they true hits?

We replaced a portion of exon 6 with a 3x FLAG sequence (region 22 – 87), given that our initial experiments with reporters containing wild-type exon 6 sequence showed protein aggregation when the reporter was transfected at a high concentration. The hits comparable to segRNA

D[10,34] target the FLAG sequence in exon 6. We have decided not to pursue those hits, since they would be irrelevant for the endogenous gene modulation.

2. Why the enrichment of segRNAs is so moderate in 4 screens?

See also our response to point #4 of reviewer #1's comments.

Initially, we considered that there was a potential technical limitation linked to the Dual-IN reporter. By RT-PCR we did observe a quite dramatic increase of exon inclusion after treatment of segRNA targeting ISS-N1. However, the increase of tdTomato in the Dual-IN reporter appears to be more moderate. We postulated that this might be because the tdTomato intensity with baseline exon 7 inclusion was relatively high (~50%) so that one could expect up to a two-fold increase which limited the dynamic range upon segRNA treatment; on the other hand, one can achieve greater reduction (as measured by fold change) upon treatment of sigRNAs. Based on this reasoning, we designed Dual-EX and performed additional screens. While we observed some improvements, particularly for segRNAs, the enhancements remained more moderate than we had expected. One plausible hypothesis is that tdTomato fluorescence might not be linear to change in exon inclusion (e.g., due to its relatively long half-life). In line with this hypothesis, we noticed that by transfection, the best sigRNA was able to reduce exon 7 inclusion to ~10%, while the best segRNA was able to increase exon 7 inclusion to ~75%. In the Dual-EX reporter, the residual tdTomato expression from ~25% exon skipping isoform might have limited the dynamic range we could achieve. The issue might be even more severe in the context of high-throughput screening where gRNAs are expressed at a lower level through low MOI, and the resulting magnitude of splicing modulation is more moderate. Our results are also consistent with several previous reports that inducing exon inclusion using dCas13d/gRNA might be inherently more challenging compared to inducing exon exclusion (Konermann et al., 2018; Leclair et al., 2020; Charles et al., 2021; Núñez-Álvarez et al., 2022), although we suggest further investigation would be required to determine if this observation is generalizable.

To address these limitations and further enhance detection limits, several technical improvements could be considered. One avenue involves augmenting gRNA expression levels, considering our observation that higher gRNA transduction correlates with an improved effect on splicing modulation. Alternatively (but not mutually exclusively), more sensitive and quantitative readout, such as deep sequencing, should also be investigated.

3. The authors should provide more detailed information about the previously identified SREs in SMN2 gene for comparison. For example, previous studies from Krainer's lab identified several splicing silencers in exon 7, intron 6, and intron 7 using ASO tiling method (Ref. 64, 65). It seems that the spliceRUSH method only identified the SREs with the strongest regulatory effects in the case of SMN2. Is this method not sensitive enough to capture the SREs with moderate effects? This issue is important for future studies on understanding of splicing regulation using the method.

We thank the reviewer for this important point. As we described in the manuscript (and responded to above), our current screen platform mainly captures large splicing changes, while more moderate effects can be missed. We agree that this is a limitation that can be improved,

but also believe that the SREs with the most dramatic regulatory effects are also the most relevant for targeted splicing modulation in the research and clinical settings. We have revised the Discussion to acknowledge the limitations of the current method:

“The current screening platform has several limitations concerning false negatives. In addition to the exonic splicing enhancer and ISS-N1, several additional splicing enhancers and silencers with weaker splicing-modulating effects were identified in previous ASO screens(Hua et al., 2007; Hua et al., 2008) but missed in our screen.”

Other points

1. In the Introduction, several descriptions are not precise.

-Page 3, line 57: RNA splicing is required not only for protein translation but also for the maturation of many lncRNAs and etc.

Corrected.

-Page 3, line 61-63: The authors did not fully describe the roles of SREs in splicing regulation. Besides recognized by splicing regulators, they have been shown to modulate splicing by forming secondary structures even through long range interactions.

Corrected. We have now rephrased our description in the introduction as follows:

“Splice site recognition and exon inclusion levels are tightly regulated via numerous splicing-regulatory elements (SREs) embedded in the exon and flanking introns. These elements can interact with hundreds of RNA-binding proteins (RBPs), or mediate the formation of local or long-range RNA-secondary structures, thereby either assisting or interfering with the recruitment of the spliceosome(Black, 2003; Licatalosi and Darnell, 2010; Ule and Blencowe, 2019).”

Descriptions throughout the text, when relevant, have also been rephrased.

2. The quality of most agarose gel images is low in the manuscript. The bands are not sharp enough, and the background of several images is relatively high (e.g. Fig. 6c). To determine splicing changes quantitatively, RT-PCR condition need to be optimized to avoid over-amplification and PCR products need to be clearly separated by agarose gel electrophoresis. Otherwise, the resulting PSI values might not be accurate for comparison.

We thank the reviewer for pointing this out. Several gel images, including Fig. 6c mentioned by this reviewer), had relatively low signals because we had to use restriction enzyme digestion to distinguish products from the endogenous *SMN1* and *SMN2* genes. We have optimized the experimental condition and obtained improved images for Fig. 6b and 6c.

3. *Fig 1a: RBPs can either enhance or inhibit the inclusion of alternative exons through binding to SREs. The arrow head in the first cartoon usually refers to “inhibitory effect”, which may cause misleading.*

We have rephrased the figure legend to avoid potential confusion:

“Schematics showing the use of dCas13d/gRNA as a programmable RNA-binding platform to map both proximal and distal SREs in their native sequence context by binding to target transcripts and competing with endogenous RNA-binding proteins (RBPs) to modulate the transcript’s splicing positively or negatively. The example illustrates that blocking of an intronic silencer induces exon inclusion.”

4. *Fig. 2g: The dark green color is hardly visible. What do the two gray rectangles indicate in the right FACS plot?*

We have tried different color schemes but agree that the difference is not obvious visually in the FACS scatter plots. To help visualize the differences, we provided the density contour plots under the FACS scatter plots, in which the difference of the two cell populations can be discerned more clearly.

The gray boxes (indicating the top 5% and bottom 5% cells regarding tdTomato/eGFP ratio) are now explained more clearly in the figure legend.

5. *Page 28, line 908: D[22,40] and E[9,30] are typos.*

The inconsistencies have been fixed.

References

- Black DL (2003) Mechanisms of alternative pre-messenger RNA splicing. *Annu Rev Biochem* 72:291-336.
- Charles EJ, Kim SE, Knott GJ, Smock D, Doudna J, Savage DF (2021) Engineering improved Cas13 effectors for targeted post-transcriptional regulation of gene expression. *bioRxiv* doi:10.1101/2021.05.26.445687.
- Hua Y, Vickers TA, Baker BF, Bennett CF, Krainer AR (2007) Enhancement of *SMN2* exon 7 inclusion by antisense oligonucleotides targeting the exon. *PLoS Biol* 5:e73.
- Hua Y, Vickers TA, Okunola HL, Bennett CF, Krainer AR (2008) Antisense masking of an hnRNP A1/A2 intronic splicing silencer corrects *SMN2* splicing in transgenic mice. *Am J Hum Genet* 82:834-848.
- Konermann S, Lotfy P, Brideau NJ, Oki J, Shokhirev MN, Hsu PD (2018) Transcriptome engineering with RNA-targeting type VI-D CRISPR effectors. *Cell* 173:665-676 e614.
- Leclair NK, Brugiolo M, Urbanski L, Lawson SC, Thakar K, Yurieva M, George J, Hinson JT, Cheng A, Graveley BR, Anczukow O (2020) Poison exon splicing regulates a coordinated network of SR protein expression during differentiation and tumorigenesis. *Mol Cell* 80:648-665 e649.
- Li SQ, Li X, Xue W, Zhang L, Yang LZ, Cao SM, Lei YN, Liu CX, Guo SK, Shan L, Wu M, Tao X, Zhang JL, Gao X, Zhang J, Wei J, Li JS, Yang L, Chen LL (2021) Screening for functional circular RNAs using the CRISPR-Cas13 system. *Nature Methods* 18:51-59.
- Licatalosi DD, Darnell RB (2010) RNA processing and its regulation: global insights into biological networks. *Nat Rev Genet* 11:75-87.
- Mendez-Mancilla A, Wessels HH, Legut M, Kadina A, Mabuchi M, Walker J, Robb GB, Holden K, Sanjana NE (2022) Chemically modified guide RNAs enhance CRISPR-Cas13 knockdown in human cells. *Cell Chem Biol* 29:321-327 e324.
- Nelson JW, Randolph PB, Shen SP, Everette KA, Chen PJ, Anzalone AV, An M, Newby GA, Chen JC, Hsu A, Liu DR (2022) Engineered pegRNAs improve prime editing efficiency. *Nat Biotechnol* 40:402-410.
- Núñez-Álvarez Y, Espie--Caullet T, Luco RF (2022) A CRISPR-dCas13 RNA-editing tool to study alternative splicing. *bioRxiv* doi: 10.1101/2022.05.24.493209.
- Singh NN, Lawler MN, Ottesen EW, Upreti D, Kaczynski JR, Singh RN (2013) An intronic structure enabled by a long-distance interaction serves as a novel target for splicing correction in spinal muscular atrophy. *Nucleic Acids Res* 41:8144-8165.
- Ule J, Blencowe BJ (2019) Alternative splicing regulatory networks: functions, mechanisms, and evolution. *Mol Cell* 76:329-345.
- Wessels H-H, Méndez-Mancilla A, Guo X, Legut M, Daniloski Z, Sanjana NE (2020) Massively parallel Cas13 screens reveal principles for guide RNA design. *Nat Biotech* 38:722-727.
- Zhang C, Konermann S, Brideau NJ, Lotfy P, Wu X, Novick SJ, Strutzenberg T, Griffin PR, Hsu PD, Lyumkis D (2018) Structural basis for the RNA-guided ribonuclease activity of CRISPR-cas13d. *Cell* 175:212-223 e217.
- Zhang Y, Nguyen TM, Zhang XO, Wang L, Phan T, Clohessy JG, Pandolfi PP (2021) Optimized RNA-targeting CRISPR/Cas13d technology outperforms shRNA in identifying functional circRNAs. *Genome Biol* 22:41.

REVIEWERS' COMMENTS

Reviewer #1 (Remarks to the Author):

In the revised version of manuscript, Recinos et al. has made clear response to our questions and performed additional experiments and discussions to strengthen the study. In this case, I believe that the manuscript is very interesting and provide a effective methods for identifying splicing-regulatory elements. I think the manuscript now is suitable for publication in Nature communications.

Reviewer #2 (Remarks to the Author):

The authors have sufficiently addressed my comments.

One minor note since Nature communications has a broad readership, in complement to screening for cis elements, it may be worthy of mentioning that dual color splicing minigene reporters have been used in the past to screen for trans factors of alternative splicing regulation in a high-throughput manner (PMID: 17060915, 23636947), so that interested readers have immediate information of these tools for deployment.

Congratulations again on this elegant study!

Reviewer #3 (Remarks to the Author):

The authors have addressed all my concerns. I have a suggestion about the title of this manuscript. I feel to include the applied technology "dCas13d/gRNA system" into the title is more accurate to describe this work. Strictly speaking, the minigene used in this study is not "a native sequence context".

Point-by-point responses to Reviewers' comments

Reviewer #1 (Remarks to the Author):

In the revised version of manuscript, Recinos et al. has made clear response to our questions and performed additional experiments and discussions to strengthen the study. In this case, I believe that the manuscript is very interesting and provide a effective methods for identifying splicing-regulatory elements. I think the manuscript now is suitable for publication in Nature communications.

We thank the reviewer for their help in improving the manuscript and the support for the publication of the work.

Reviewer #2 (Remarks to the Author):

The authors have sufficiently addressed my comments.

One minor note since Nature communications has a broad readership, in complement to screening for cis elements, it may be worthy of mentioning that dual color splicing minigene reporters have been used in the past to screen for trans factors of alternative splicing regulation in a high-throughput manner (PMID: 17060915, 23636947), so that interested readers have immediate information of these tools for deployment.

Congratulations again on this elegant study!

We thank the reviewer for their help in improving the manuscript and the support for the publication of the work. We apologize for the omission of relevant references, and have included them as suggested (p 5):

“We note that similar fluorescent splicing reporters were used previously for high-throughput screening of splicing regulators or splicing-disrupting mutations^{21,66,67.}”

Reviewer #3 (Remarks to the Author):

The authors have addressed all my concerns. I have a suggestion about the title of this manuscript. I feel to include the applied technology “dCas13d/gRNA system” into the title is more accurate to describe this work. Strictly speaking, the minigene used in this study is not “a native sequence context”.

We thank the reviewer for their help in improving the manuscript and the support for the publication of the work. We have also changed the title following the reviewer's suggestion:

“CRISPR-dCas13d based deep screening of proximal and distal splicing-regulatory elements”.